# Sam68 promotes self-renewal and glycolytic metabolism in mouse neural progenitor cells by modulating *Aldh1a3* pre-mRNA 3'-end processing

Piergiorgio La Rosa[1,2], Pamela Bielli[1,2], Claudia Compagnucci[1,2], Eleonora Cesari[1,2], Elisabetta Volpe[3], Stefano Farioli Vecchioli[4], Claudio Sette[1,2]*

[1]Department of Biomedicine and Prevention, University of Rome Tor Vergata, Rome, Italy; [2]Laboratory of Neuroembryology, Fondazione Santa Lucia, Rome, Italy; [3]Laboratory of Neuroimmunology, Fondazione Santa Lucia, Rome, Italy; [4]Institute of Cell Biology and Neurobiology CNR, Rome, Italy

**Abstract** The balance between self-renewal and differentiation of neural progenitor cells (NPCs) dictates neurogenesis and proper brain development. We found that the RNA-binding protein Sam68 (Khdrbs1) is strongly expressed in neurogenic areas of the neocortex and supports the self-renewing potential of mouse NPCs. Knockout of *Khdrbs1* constricted the pool of proliferating NPCs by accelerating their cell cycle exit and differentiation into post-mitotic neurons. Sam68 function was linked to regulation of *Aldh1a3* pre-mRNA 3'-end processing. Binding of Sam68 to an intronic polyadenylation site prevents its recognition and premature transcript termination, favoring expression of a functional enzyme. The lower ALDH1A3 expression and activity in *Khdrbs1*[-/-] NPCs results in reduced glycolysis and clonogenicity, thus depleting the embryonic NPC pool and limiting cortical expansion. Our study identifies Sam68 as a key regulator of NPC self-renewal and establishes a novel link between modulation of ALDH1A3 expression and maintenance of high glycolytic metabolism in the developing cortex.

*For correspondence: claudio.sette@uniroma2.it

**Competing interests:** The authors declare that no competing interests exist.

## Introduction

Neurogenesis is the process that leads to a regionalized brain from an embryonic neuroepithelium. In the mouse developing brain, most neurons are generated between the 10th day of embryonal life (E10) and birth (*Martynoga et al., 2012*), even though adult neurogenesis persists in restricted areas near the lateral ventricles, the ventricular (VZ) and subventricular (SVZ) zones (*Bjornsson et al., 2015*), and in the subgranular zone of the dentate gyrus of the hippocampus (*Urbán and Guillemot, 2014*). Neurogenesis is fueled by specialized stem cells that are collectively named Neural Stem/Progenitor Cells (NPCs) (*Johansson et al., 2010*; *Taverna et al., 2014*). As embryonic neurogenesis starts, neuroepithelial cells at first, and radial glia cells (RGCs) later, divide symmetrically in the VZ, to generate cells retaining self-renewal capacity, or asymetrically to give rise to intermediate progenitor cells (IPCs) or differentiated cells (*Paridaen and Huttner, 2014*; *Taverna et al., 2014*). IPCs divide away from the VZ, one or more times, before they generate post-mitotic neurons, which localize in the basal compartment of the developing cortex (cortical layer). These divisions of neural precursors need to be tightly modulated, as proper regulation of neurogenesis is of paramount importance to achieve the right number of neurons and the correct expansion and stratification of the cortex during development. In particular, fine-tuned control of NPC fate is required to balance self-renewing divisions with asymmetric divisions (*Taverna et al., 2014*; *Fernández et al., 2016*).

**eLife digest** Neurons develop from cells called neural progenitors. These cells can either divide to produce more progenitor cells or develop into specific types of neurons. These two activities – known as self-renewal and differentiation – must be balanced to produce the right number of specialized neurons, without depleting the pool of progenitor cells. The self-renewal and differentiation of progenitor cells is balanced by essentially regulating which genes are active, or expressed, within the cells.

In the first step of gene expression, the genetic instructions are copied to form a molecule of pre-messenger RNA (or pre-mRNA for short). Each pre-mRNA molecule is then processed to produce a final product that can be translated into protein. Importantly, two copies of the same pre-mRNA may sometimes be processed in different ways, which allows multiple proteins to be produced from a single gene.

RNA-binding proteins control pre-mRNA processing. The expression of one such protein, called Sam68, oscillates during the development of the nervous system, such that its expression peaks when there is intense production of new neurons and then declines. However, it was not known whether Sam68 actually helps neurons to develop.

La Rosa et al. have now analysed the role of Sam68 in the developing brain of mice. The experiments confirmed that Sam68 is highly expressed in neural progenitor cells and showed that its levels dictate the cell's fate: high expression encourages a cell to self-renew, while low expression triggers it to develop into a specialized neuron.

Further investigation revealed that Sam68 works by promoting the expression of a metabolic enzyme called Aldehyde Dehydrogenase 1A3 or ALDH1A3. This enzyme promotes the release of energy from molecules of glucose via a process known as anaerobic glycolysis. La Rosa et al. found that cells that lack Sam68 make a truncated version of the pre-mRNA encoding ALDH1A3. This truncated pre-mRNA encodes a shortened version of the enzyme that is inactive. Further experiments confirmed that Sam68 normally prevents this from happening by binding to the pre-mRNA and processing it to produce the full-length, working version of the ALDH1A3 enzyme. Also, La Rosa et al. found that progenitor cells need working ALDH1A3 to keep them dividing, and to stop them from developing into specialized neurons too soon.

Finally, because the processing of pre-RNA plays a major role in brain development, problems with this process often lead to intellectual disabilities and neurodegenerative diseases, such as autism spectrum disorder and amyotrophic lateral sclerosis. The next step following on from these new findings will be to investigate whether defects in Sam68 contribute to such conditions and, if so, to look for ways to counteract these defects.

Conversely, dysregulation of neurogenesis can result in developmental disorders linked to neurological pathologies (*Sun and Hevner, 2014*). Thus, understanding the molecular mechanisms that control neurogenesis may pave the path to novel approaches to these diseases.

Multiple mechanisms, including regulation of metabolic routes (*Shyh-Chang et al., 2013*) and of gene expression (*Paridaen and Huttner, 2014*), cooperate to create an interconnected network that balances NPC self-renewal and differentiation in the correct time and space during cortical expansion. Mounting evidence documents that modulation of RNA metabolism, and particularly of alternative splicing, plays a key role in neurogenesis and during formation of neuronal circuits, as highlighted by the global changes in the splicing signature that accompany the transition from NPCs to neurons (*Zheng and Black, 2013*; *Raj and Blencowe, 2015*). Splicing is operated by the spliceosome, a large ribonucleoprotein machinery that mediates sequentially ordered reactions to excise introns and ligate exons (*Wahl et al., 2009*). Through regulated assortment of multiple exons and introns during pre-mRNA processing, alternative splicing allows the production of many splice variants from most genes, thus greatly amplifying the coding potential and plasticity of the genome (*Fu and Ares, 2014*). Notably, brain is among the tissues displaying the largest extent of alternative splicing, which likely contributes to the complexity of neural circuits (*Raj and Blencowe, 2015*). In support of this notion, several splicing factors have been shown to play key roles during

neurogenesis and/or specific brain functions (*Raj and Blencowe, 2015*). For instance, a temporal switch in the expression of two homologous polypyrimidine-tract-binding proteins, PTBP1 and PTBP2, governs splicing of a large set of neural-specific exons in genes involved in neuronal functions (*Boutz et al., 2007*). Knockout of the *Ptbp2* gene caused premature neurogenesis and depletion of the NPC pool (*Licatalosi et al., 2012*), proving the crucial role played by this splicing factor in the developing brain. Similarly, the neural-specific serine-arginine (SR)-related protein of 100 kDa (nSR100) regulates a network of exons in genes involved in neuronal functions and knockout of this gene in mice leads to widespread neurodevelopmental defects (*Calarco et al., 2009*; *Quesnel-Vallières et al., 2015*). Another splicing factor involved in neuronal functions is Sam68, encoded by the *Khdrbs1* gene, which is highly expressed in brain and testis (*Richard et al., 2005*; *Paronetto et al., 2009*), and it was shown to be involved in the pathogenesis of fragile X tremor/ataxia syndrome (*Sellier et al., 2010*) and spinal muscular atrophy (*Pedrotti et al., 2010*; *Pagliarini et al., 2015*). Furthermore, Sam68 modulates splicing of the neurexin one gene (*Nrxn1*) in response to neuronal activity (*Iijima et al., 2011*) and its ablation caused altered synaptic plasticity and motor coordination defects (*Lukong and Richard, 2008*; *Iijima et al., 2011*).

In this study, we have investigated the role of Sam68 during the development of the central nervous system. Sam68 is strongly expressed during cortical expansion (E10.5–15.5), whereas its levels decline after birth. We found that Sam68 regulates the switch between self-renewal and differentiation of mouse NPCs both in vivo and in vitro. Splicing-sensitive microarrays identified a subset of genes and exons whose expression is dependent on Sam68 in NPCs. In particular, Sam68 prevents usage of a cryptic polyadenylation signal (PAS) in intron 7 of the Aldehyde Dehydrogenase 1A3 (*Aldh1a3*) gene, thus promoting the expression of a functional enzyme. We also found that ALDH1A3 enhances anaerobic glycolytic metabolism in NPCs and that enforcing its activity rescued the self-renewal defect of Sam68 knockout (*Khdrbs1*$^{-/-}$) NPCs. Conversely, *Aldh1a3* knockdown in wild-type cells mimicked the phenotype of Sam68 knockout NPCs, by reducing glycolytic activity and promoting neuronal differentiation. Thus, our work unveils a key role of Sam68 in neurogenesis through regulation of *Aldh1a3* pre-mRNA processing, which results in the modulation of glycolytic metabolism and NPC fate during cortical development.

## Results

### Sam68 is highly expressed in neurogenic cells of the developing cortex

Sam68 is a KH-domain RNA-binding protein involved in several steps of RNA metabolism (*Bielli et al., 2011*; *Frisone et al., 2015*). Developmental analysis of the mouse cortex showed that Sam68 mRNA and protein levels peak between E13.5 and E15.5, whereas its expression slowly declines thereafter and is minimal from 9 days post-partum (9dpp) until adulthood (*Figure 1A,B*). The peak of Sam68 expression corresponds to stages of intense neurogenesis in the developing cortex (*Paridaen and Huttner, 2014*; *Taverna et al., 2014*) and parallels that of the NPC marker SOX2, which is also high between E10.5 and E15.5 and sharply decreases in post-natal stages (*Figure 1B*). Furthermore, Sam68 is strongly expressed in neurogenic periventricular regions of E13.5 brain, like SOX2 (*Figure 1C*). Sam68 and SOX2 co-localized in most cells of the VZ and SVZ of E13.5 cortex (*Figure 1D*), and their expression was even more restricted to these cortical zones at 1dpp (*Figure 1E*). These results suggested that Sam68 expression is high in NPCs and declines upon differentiation. To test this hypothesis, NPCs were isolated from E13.5 cortex and cultured in vitro under proliferating or differentiating conditions (*Bertram et al., 2012*). Sam68, like SOX2, was abundant in proliferating NPC (0d) and steadily decreased when cells were induced to differentiate (1d-6d in *Figure 1F,G*). Conversely, expression of the neuronal marker TUBB3 (βIII-tubulin) was barely detectable in proliferating NPCs and augmented upon differentiation (*Figure 1F,G*). Thus, Sam68 is highly expressed in embryonic NPCs.

### Ablation of Sam68 function accelerates differentiation of NPCs in the embryonic cortex

To investigate whether Sam68 plays a role during neurogenesis, we first assessed whether its ablation affected the rate of NPC proliferation in the embryonic cortex. To this end, proliferating cells were labeled by injecting BrdU in pregnant females at 13.5 days post-coitum (dpc) 2 hr before

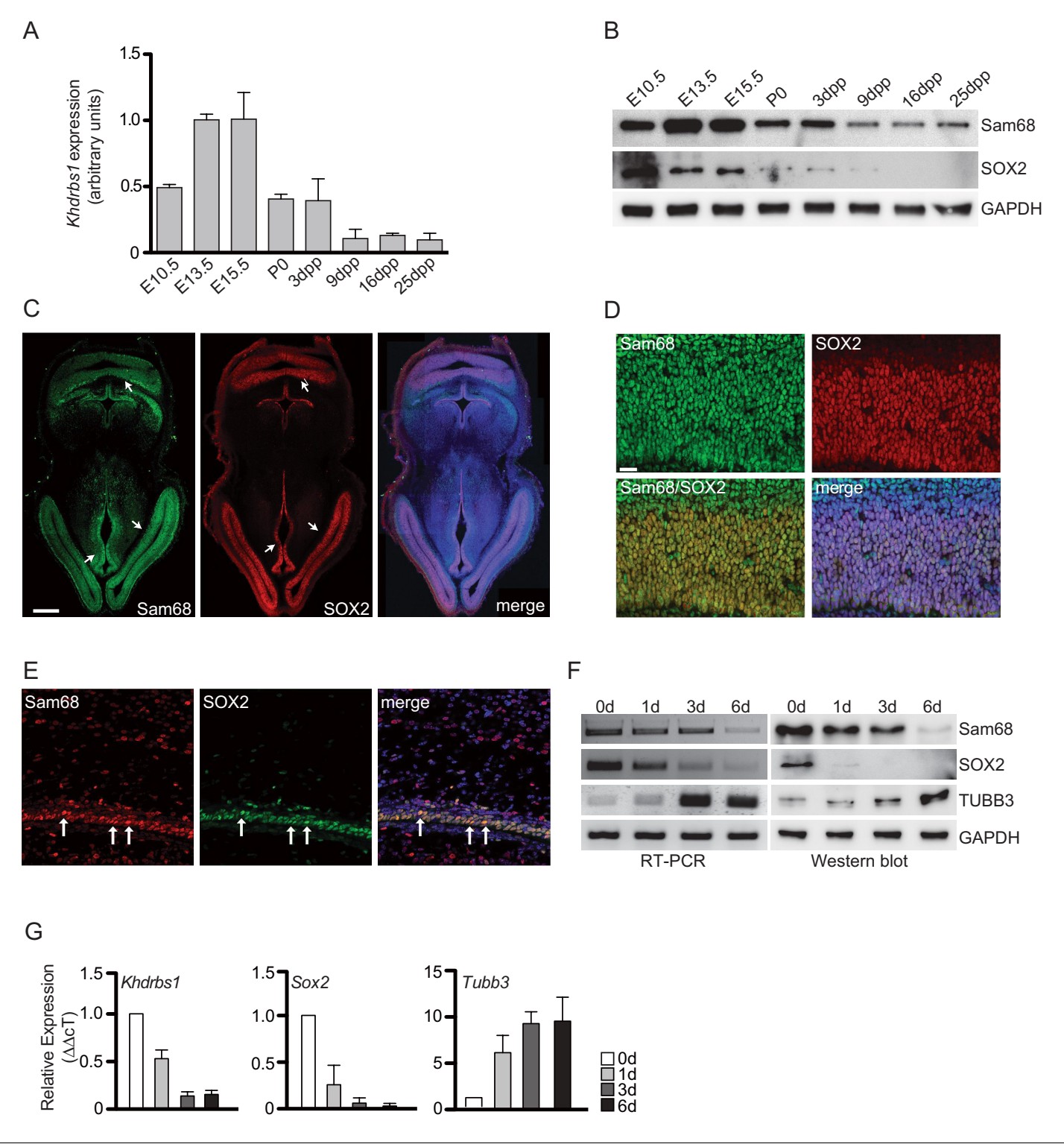

**Figure 1.** Sam68 is highly expressed in NPCs and decreases during differentiation. (**A**) qPCR analysis of *Khdrbs1* mRNA levels in the cortex of embryonic (E10.5-E15.5) and post-natal (P0-25dpp) mouse brain. *Khdrbs1* relative expression was evaluated by △CT method using *L34* expression for normalization. (**B**) Western blot analysis of Sam68 and SOX2 expression in lysates from embryonic (E10.5-E15.5) and post-natal (P0-25dpp) mouse cortices. GAPDH was used as loading control. (**C** and **D**) Immunofluorescence analyses of Sam68 and SOX2 expression in E13.5 mouse brain. (**C**) Horizontal sections of whole brain; white arrows point to periventricular zones where both proteins are highly expressed. Scale bar = 250 µm. (**D**) High-magnification confocal images confirm Sam68 and SOX2 colocalization in most cells of the VZ and SVZ. Scale bar = 25 µm. (**E**) High magnification of

*Figure 1 continued on next page*

*Figure 1 continued*

confocal images of 1 dpp mouse VZ-SVZ, show the colocalization of Sam68 and SOX2 in NPCs (white arrows). Scale bar = 25 µM. (F) Analysis of Sam68, SOX2 and TUBB3 mRNA (left panels) and protein levels (right panels) in NPCs cultured under proliferating condition (0d) or during 1–6 days of differentiation (1-6d). (G) qPCR analysis of *Khdrbs1, Sox2 and Tubb3* mRNA levels in NPCs under proliferation conditions (0d) and 1–6 days of differentiation (1, 3, 6d). Relative expression was evaluated using △△CT method and 0d as reference point. *L34* expression was used for the initial △CT normalization. NPCs, Neural progenitor cells.

embryo (E13.5) collection. Proliferating cells (BrdU-positive cells) were significantly decreased in the cortex of *Khdrbs1*[-/-] embryos (*Figure 2—figure supplement 1A*). Likewise, SOX2[+]-NPCs, as well as the number of NPCs undergoing DNA synthesis (SOX2[+]/BrdU[+] cells), were also markedly affected (*Figure 2—figure supplement 1A*), indicating that fewer NPCs are present and proliferate in the E13.5 *Khdrbs1*[-/-] cortex.

At this developmental stage, two main types of proliferating NPCs have been described in the mouse cortex (*Paridaen and Huttner, 2014*; *Taverna et al., 2014*): RGCs and IPCs. RGCs express the transcription factor PAX6 and divide in the VZ and SVZ (apical layers). IPCs express the transcription factor TBR2, are derived from RGCs and divide one or more times away from the VZ before they generate post-mitotic neurons, which localize in the basal compartment of the cortex (cortical layer) and express the transcription factor TBR1 (*Englund et al., 2005*). Thus, these three markers are expressed in sequentially ordered manner from the apical (PAX6) to the basal (TBR1) layer of the E13.5 neuroepithelium (*Figure 2A*). Strikingly, total (PAX6[+] cells) and proliferating RGCs (PAX6[+]/BrdU[+] cells) and IPCs (TBR2[+] and TBR2[+]/BrdU[+] cells) were markedly reduced in the knockout cortex (*Figure 2B,C*), whereas differentiated neurons (TBR1[+] cells) were almost doubled (*Figure 2D*). No TBR1-positive cell was labeled by BrdU in both wild-type and knockout embryos (*Figure 2—figure supplement 1B*), confirming that these cells are post-mitotic neurons (*Englund et al., 2005*).

The concomitant decrease in proliferating cells and increase in post-mitotic neurons observed in the knockout embryos suggested that, in the absence of Sam68, NPCs are prone to exit the cell cycle and differentiate. To directly test this hypothesis, we injected pregnant females at 12.5dpc with BrdU and collected the embryos 48 hr later (E14.5) (*Figure 3A*). Co-staining with the Ki67 proliferation marker revealed that the fraction of NPCs that exited the cell cycle (Ki67[-]/BrdU[+]/ BrdU[+] cells) in the E12.5-E14.5 time-frame was significantly increased in *Khdrbs1*[-/-] embryos (*Figure 3B*). Furthermore, in wild-type embryos, ~30% of post-mitotic neurons were also labeled with BrdU (*Figure 3C*), indicating that only a subset of NPCs underwent terminal differentiation within 48 hr. By contrast, this process was strongly enhanced in knockout embryos, with more than 80% of TBR1-positive cells that were labeled with BrdU (*Figure 3C*). These results suggest that Sam68 is required to promote NPC self-renewal and to delay their differentiation into post-mitotic neurons in the mouse embryonic cortex.

To determine if premature exit from the cell cycle and differentiation of NPCs has an effect on cortical expansion, we examined embryos at E17.5, when neurogenesis is almost completed (*Englund et al., 2005*). A 2-hr pulse showed that BrdU-positive proliferating cells were strongly decreased in *Khdrbs1*[-/-] cortex also at this stage, which was reflected in lower number of SOX2-positive NPCs (*Figure 3D,E*). Furthermore, although they were indistinguishable in size from the wild type and heterozygote littermates (*Figure 3—figure supplement 1*), cortical expansion, measured as length from ventricular to pial surface, resulted significantly reduced in E17.5 *Khdrbs1*[-/-] embryos (*Figure 3D,E*). These observations strongly suggest that ablation of Sam68 depletes the NPC pool and limits cortical expansion during neurogenesis.

## Sam68 promotes self-renewal and clonogenic potential of NPCs

To elucidate the function of Sam68 in NPCs, we isolated them from wild type and knockout E13.5 cortices (*Bertram et al., 2012*). NPCs grown under stemness conditions form characteristic neurospheres that rapidly grow in volume. We observed that knockout neurospheres grew at slower rate than wild type ones (*Figure 4—figure supplement 1A*), even though no differences were detected in the extent of cell death (*Figure 4—figure supplement 1B*), and appeared irregular in shape with a tendency to form protrusions, due to cells that attached to the substrate and exited the sphere (*Figure 4A*). Furthermore, when seeded at single cell level, knockout NPCs produced cytoplasmic

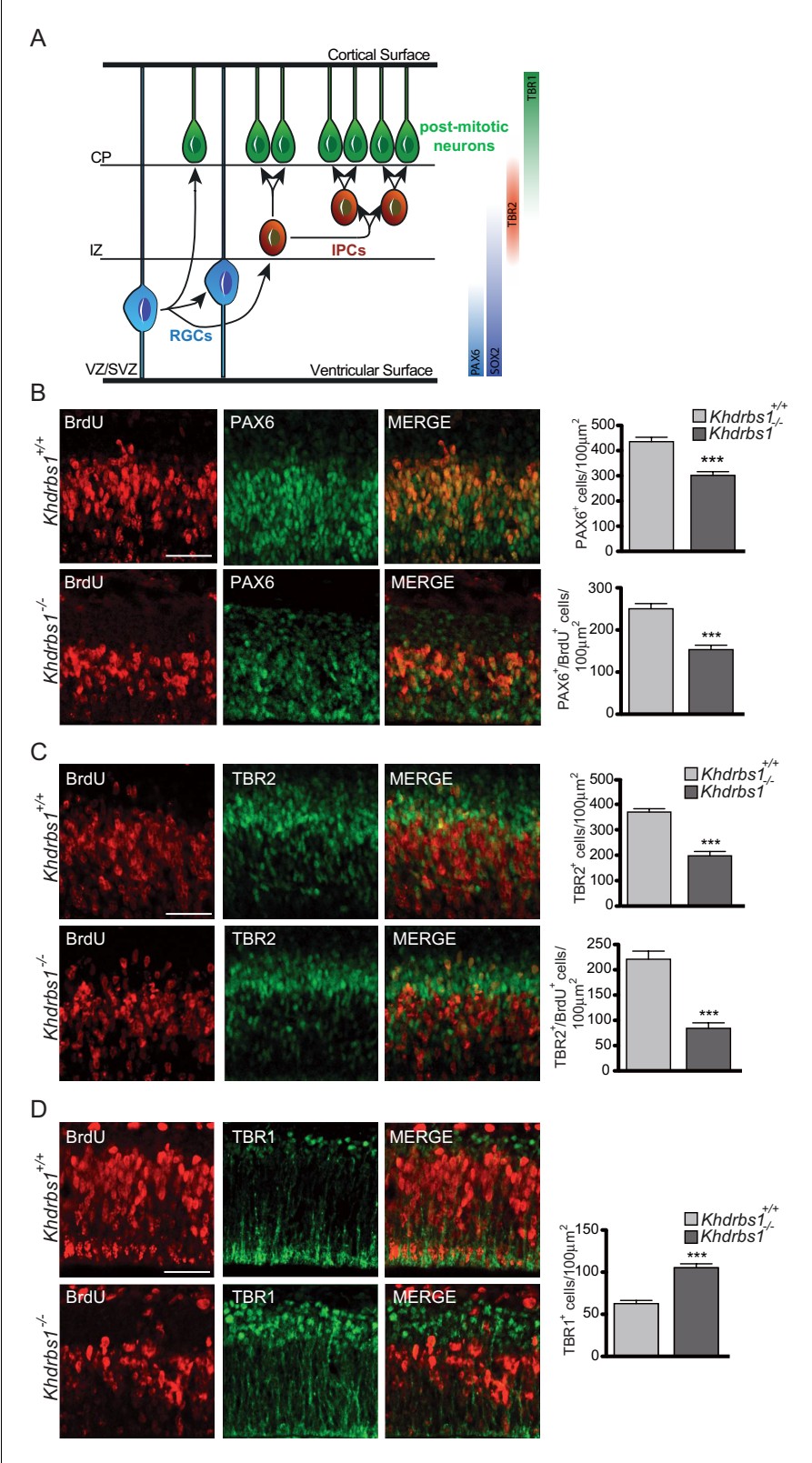

**Figure 2.** Ablation of *Khdrbs1* perturbs neurogenesis in the embryonic cortex. (**A**) Schematic representation of the stratification of the VZ/SVZ, intermediate zone (IZ) and cortical plate (CP) in the E13.5 embryonic cortex. Molecular markers, expressed by the respective cell types are represented on the right side of the scheme. (**B–D**) Immunofluorescence analyses on sections of *Khdrbs1*$^{+/+}$ and *Khdrbs1*$^{-/-}$ brain from E13.5 embryos treated for 2 hr with BrdU to label proliferating cells. Sections were stained with anti-PAX6 (**B**), anti-TBR2 (**C**) or anti-TBR1 (**D**). All sections were co-stained with anti-
*Figure 2 continued on next page*

*Figure 2 continued*

BrdU antibody. Bar graphs on the right side show the number of BrdU/PAX6 (B), BrdU/TBR2 (C) and TBR1 (D) positive cells counted in 100 µm². N = 3; ***p<0.001. Scale bar = 100 µm. SVZ, Subventricular Zone; VZ, Ventricular Zone; IZ, Intermediate Zone; CP, Cortical Plate; RGCs, Radial Glia Cells; IPCs, Intermediate Progenitor Cells.

The following figure supplement is available for figure 2:

**Figure supplement 1.** Ablation of *Khdrbs1* perturbs neurogenesis in the embryonic cortex.

extensions and attached to the plate, while wild-type cells remained in suspension (*Figure 4—figure supplement 1C*). Collectively, these features suggested that *Khdrbs1*⁻/⁻ NPCs display a tendency to lose stemness and differentiate. To directly test this possibility, we performed clonogenic assays at different passages from NPC isolation. The clonogenic potential of wild-type NPCs slowly declined from ~45% to~25% after 7 passages in culture. However, loss of clonogenicity was markedly accelerated in knockout NPCs, reaching ~10% by passage 7 (*Figure 4B*).

Next, we performed differentiation assays by switching NPCs to serum-containing medium for 1–3 days (*Bertram et al., 2012*; *Compagnucci et al., 2013*). *Khdrbs1*⁻/⁻ NPCs differentiated with higher efficiency than wild-type cells, as indicated by the fewer cells that retained high expression of the stemness markers PAX6 and Nestin after 1 day (*Figure 4C,D*). When differentiation was evaluated after 3 days, *Khdrbs1*⁻/⁻ NPCs gave rise to significantly more neurons and oligodendrocytes than the corresponding wild-type NPCs (*Figure 4E,F*). Furthermore, neurons generated by knockout NPCs exhibited higher complexity and more advanced differentiation stage, as documented by their increased branching level (*Figure 4E*, *Figure 4—figure supplement 2*). These findings indicate that *Khdrbs1*⁻/⁻ NPCs have reduced self-renewal proficiency and enhanced tendency to differentiate both in vivo and in vitro.

## Sam68 regulates the expression and splicing of genes involved in differentiation of neurons and glial cells

We hypothesized that the reduced self-renewal capacity of *Khdrbs1*⁻/⁻ NPCs was due to dysregulation of gene expression. Splicing-sensitive exon arrays identified 254 annotated genes regulated at the expression levels and 117 exons in 87 genes regulated at the splicing level in knockout NPCs (*Figure 4—figure supplement 3A*). Sam68 generally acted as a repressor of expression and the majority of targets were upregulated in knockout NPCs (*Figure 4—figure supplement 3A*). By contrast, Sam68 functioned as splicing enhancer for target exons, with little overlap between genes regulated at expression and splicing level (*Figure 4—figure supplement 3A,B*). Exon cassette was the predominant splicing event regulated by Sam68 in NPCs, followed by alternative last exon selection (*Figure 4—figure supplement 3C*). Notably, gene ontology analysis highlighted significant enrichment of functional categories related to neuron and oligodendrocyte properties (cluster 1 in *Figure 4—figure supplement 3D*), thus supporting the pro-differentiation effect of Sam68 depletion in NPCs. For instance, genes encoding neuronal proteins like FXYD, a transmembrane modulator of the Na, K-ATPase enzyme linked to RETT syndrome (*Deng et al., 2007*), and DDC, the DOPA decarboxilase that catalyzes dopamine production (*Bertoldi, 2014*), were upregulated (*Supplementary file 1*). Similarly, the oligodendrocyte-specific myelin-associated glycoprotein (MAG) and myelin basic protein (MBP) were strongly induced in *Khdrbs1*⁻/⁻ NPCs (*Figure 4—figure supplement 3E*; *Figure 4-source data file 1*). These results confirmed at the molecular level the pro-differentiation phenotype exhibited by *Khdrbs1*⁻/⁻ NPCs.

## Sam68 directly modulates alternative 3′-end processing of *Aldh1a3* in NPCs

Sam68 is mainly known for its direct role in the regulation of pre-mRNA processing (*Frisone et al., 2015*). Thus, we further investigated the functional relevance of splicing-regulated genes to identify targets that could cause the phenotype of knockout NPCs. Among the 87 genes identified in the array (*Figure 4—figure supplement 3B*; *Figure 4-source data file 2*), we focused on *Aldh1a3* for its potential relevance in NPCs. Indeed, ALDH1A3 was shown to support self-renewal and clonogenic

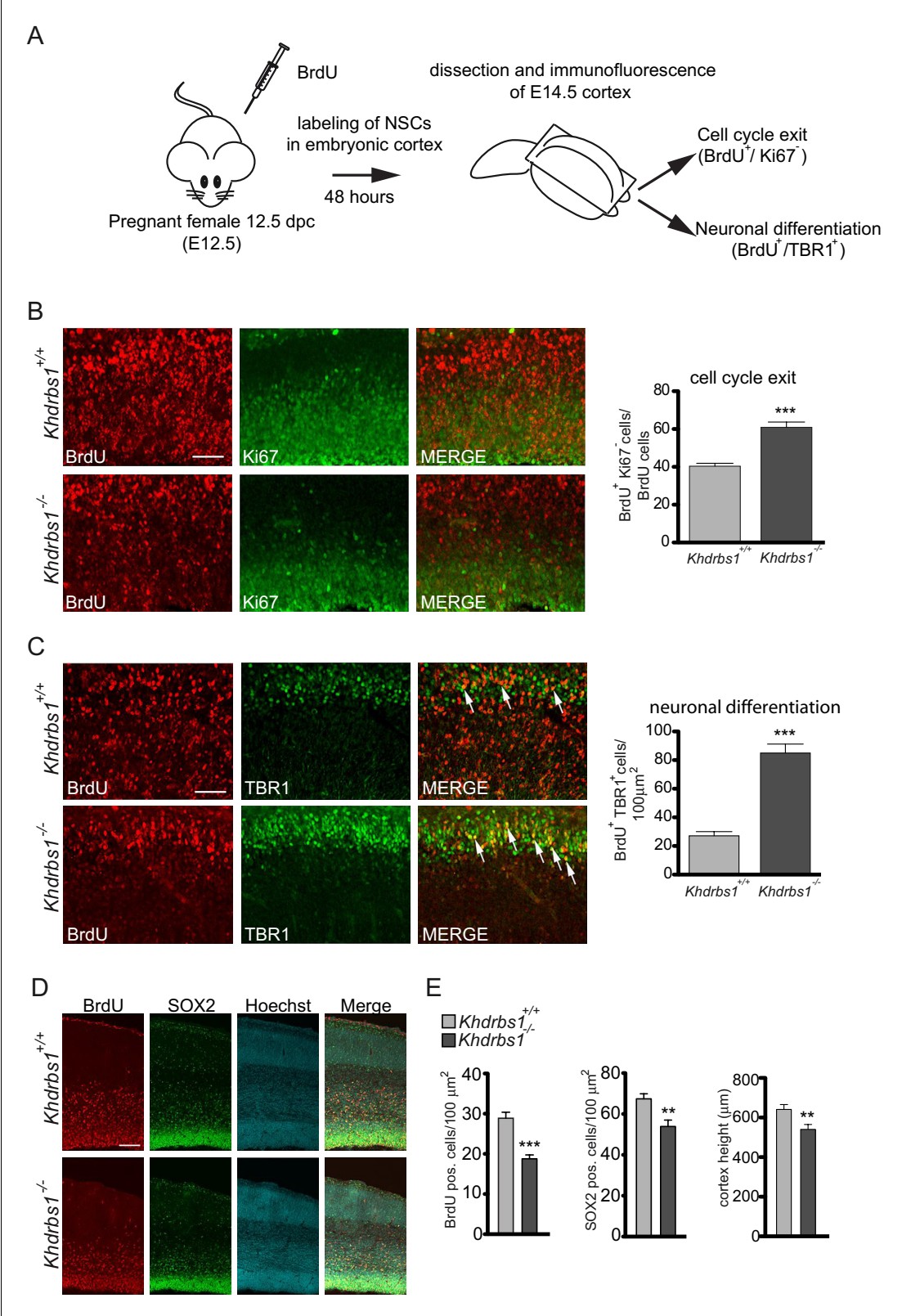

**Figure 3.** Differentiation of NPCs in post-mitotic neurons is accelerated in the *Khdrbs1⁻/⁻* mouse cortex. (**A**) Diagram of the experimental design to investigate the fate on NPCs during neurogenesis. Pregnant females were treated with BrdU at E12.5, and embryos were collected after 2 days for immunofluorescence analyses. (**B** and **C**) Confocal images of immunofluorescence analysis of Ki67 (**B**) and TBR1 (**C**) in *Khdrbs1⁺/⁺* and *Khdrbs1⁻/⁻* E14.5 cortex. All sections were co-stained with anti-BrdU antibody. Bar graphs on the right side show the number of $BrdU^+/Ki67^-$ cells (**B**) and of $BrdU^+/TBR1^+$
*Figure 3 continued on next page*

*Figure 3 continued*

cells (**C**) counted in 100 µm². N = 3; ***p<0.001; scale bar = 100 µm. (**D**) Representative images of embryonic cortex of *Khdrbs1*⁺/⁺ and*Khdrbs1*⁻/⁻ E17.5 embryos treated with BrdU 2 hr before collection. Sections were processed for immunofluorescence analysis of BrdU and SOX2 and co-stained with Hoechst to detect nuclei. (**E**) Bar graphs represent (mean±SD) measurement of cortex length from ventricular to pial surface (left), number of BrdU⁺ cells (middle) and SOX2⁺ cells (right). N = 3; *** p<0.001; ** p<0.01. Scale bar = 100 µm.

The following figure supplement is available for figure 3:

**Figure supplement 1.** Ablation of *Khdrbs1* does not affect overall embryo development.

potential of cancer stem cells (*Duan et al., 2016*), including glioma stem cells (*Mao et al., 2013*), which can differentiate into neurons and glial cells like NPCs (*Lathia et al., 2015*).

The 5' region of the *Aldh1a3* transcript (exons 1–7) was upregulated in *Khdrbs1*⁻/⁻ NPCs, whereas the 3' region (exons 8–13) was downregulated (*Figure 5A,B*, *Figure 5—figure supplement 1A*). Closer inspection of the array results highlighted the potential upregulation of a cryptic alternative last exon (ALE) residing in the proximal part of intron 7, suggesting the existence of an alternative PAS in this intron and premature termination of the transcript (*Figure 5A*). To verify this hypothesis, we performed 3'-RACE experiments using an oligo-dT anchor as reverse primer and forward primers located in the 5' region of intron 7 (primer 1) and at the exon-intron junction (primer 2) (*Figure 5C*). Both primers amplified a distinct band from *Khdrbs1*⁻/⁻ NPCs RNA, whereas only faint bands of various sizes were detected in wild type cells (*Figure 5D*). Sequencing of the 3'-RACE products amplified in knockout NPCs revealed usage of an alternative PAS in intron 7 that is followed by two potential cleavage sites (*Figure 5E*). To confirm this result, and to rule out accumulation of intron 7-containing pre-mRNA, we performed RT-PCR analyses using a forward primer in exon 6 and reverse primers located either upstream (primers 1 and 2) or downstream (primer 3) of the alternative PAS and cleavage sites (*Figure 5—figure supplement 1B*). A spliced transcript containing intron 7 sequences could be amplified only with the primers situated upstream of the alternative PAS (*Figure 5—figure supplement 1B*). Furthermore, incubation of *Khdrbs1*⁻/⁻ NPCs with 5,6-dichloro-1-β-D-ribofuranosylbenzimidazole (DRB) to block RNA transcription led to rapid decay of transcripts containing exon 3-intron 3 and intron 8-exon 8 sequences, indicative of pre-mRNA amplification, whereas the amplicon corresponding to exon 7-intron 7 sequences was stable up to 6 hr of incubation (*Figure 5—figure supplement 1C*). These results confirm the presence of an alternative mRNA originating from premature termination of the *Aldh1a3* transcript in knockout NPCs.

The canonical PAS sequences (AAUAAA and AUUAAA) correspond to optimal binding sites for Sam68 (*Ray et al., 2013*; *Feracci et al., 2016*). We hypothesized that Sam68 can bind the cryptic PAS in intron 7 in wild-type NPCs, thus preventing its recognition and blocking premature termination of the pre-mRNA. In line with this possibility, UV crosslink immunoprecipitation (CLIP) experiments showed strong recruitment of Sam68 in proximity of the cryptic PAS in intron 7 of the *Aldh1a3* pre-mRNA, whereas binding in other regions was much weaker (*Figure 5F*). These observations strongly suggest that Sam68 binding is required to mask the cryptic PAS and avoid premature termination of the *Aldh1a3* transcript in mouse NPCs.

## ALDH1A3 protein expression and activity are reduced in *Khdrbs1*⁻/⁻ NPCs

Alternative PAS usage should lead to lower expression of the full-length ALDH1A3 protein and potential expression of a truncated ALDH1A3 protein. Western blot analysis confirmed the reduction of the full-length ALDH1A3 protein (*Figure 6A*). However, we did not detect a band corresponding to the truncated protein (ALDH1A3△) originating from the ALE in intron 7 (*Figure 6—figure supplement 1A*). Blocking the proteasome by incubation with MG132 did not lead to ALDH1A3△ accumulation, whereas interference with autophagy using chloroquine caused a mild accumulation of a protein product slightly smaller than the recombinant ALDH1A3△ (*Figure 6—figure supplement 1A*). Thus, ALDH1A3△ appears to be either very unstable or not expressed in NPCs. Nevertheless, to verify if such truncated protein could be functional, we evaluated its enzymatic activity. Full-length ALDH1A3 overexpressed in HEK293 exhibited enzymatic activity, as measured by reduction of NADPH (*Lindahl et al., 1982*), whereas the ALDH1A3△ protein was completely inactive (*Figure 6—*

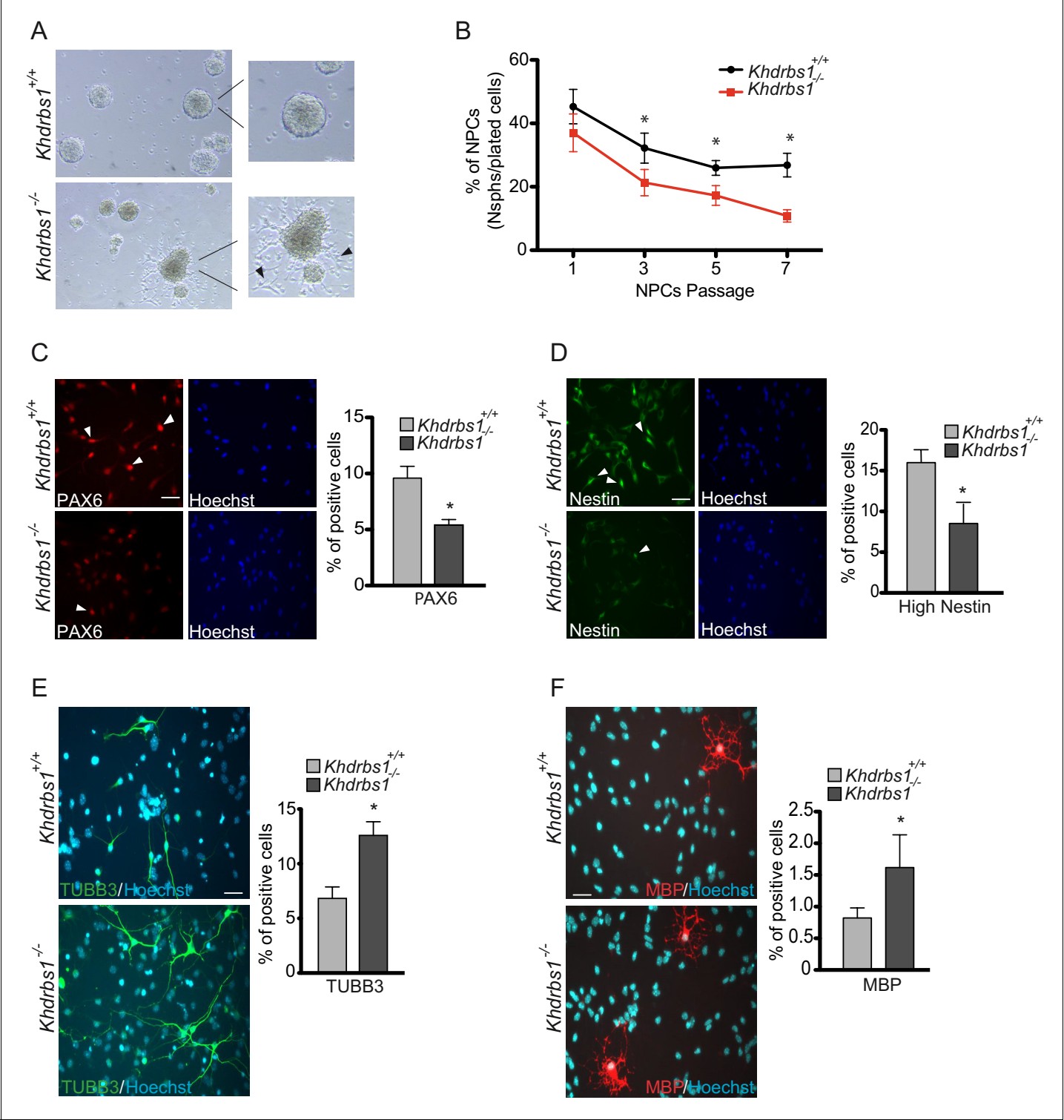

**Figure 4.** *Sam68*[-/-] NPCs lose stemness and are prone to differentiate in culture. (**A**) Bright field images of *Khdrbs1*[+/+] and *Khdrbs1*[-/-] NPCs cultured in proliferating condition for 3 days in culture. The insets show higher magnification of a selected neurosphere. Black arrowheads point to *Khdrbs1*[-/-] NPCs exiting the sphere and attaching to the surface. (**B**) Clonogenic assay of *Khdrbs1*[+/+] and *Khdrbs1*[-/-] NPCs. Clonogenicity was expressed as the percentage of neurospheres obtained from the seeded NPCs at each indicated passage. Data represent mean±SD of 3 independent experiments; * $p<0.05$. (**C** and **D**) Immunofluorescence analysis of the stemness markers PAX6 (**C**) and Nestin (**D**) in *Khdrbs1*[+/+] and *Khdrbs1*[-/-] NPCs cultured in differentiating condition for 1 day. Arrowheads point to high-PAX6[+] cells (**C**) and high-Nestin[+] cells (**D**). Bar graphs represent (mean±SD) measurement of number of high-PAX6[+] cells (**C**) and high-Nestin[+] cells (**D**). N = 3; * $p<0.05$; scale bars 50 μm. (**E** and **F**) Immunofluorescence analysis of expression of

*Figure 4 continued on next page*

*Figure 4 continued*

the neuronal marker TUBB3 (**E**) and of the oligodendrocyte marker MBP (**F**) in *Khdrbs1⁺/⁺* and *Khdrbs1⁻/⁻* NPCs, differentiated for 3 days. Bar graphs represent (mean±SD) measurement of number of TUBB3⁺ cells (**E**) and MBP⁺ cells (**F**). Analysis results were reported on the graphs on the right of the images. N = 3; * p<0.05; scale bars 50 µm. NPCs, Neural progenitor cells.

The following source data and figure supplements are available for figure 4:

**Source data 1.** List of the genes upregulated and downregulated inSam68 knockout NPCs.

**Source data 2.** List of exons regulated by SAM68 in mouse NPCs.

**Figure supplement 1.** *Khdrbs1⁻/⁻* NPCs lose stemness and are prone to differentiate in culture.

**Figure supplement 2.** *Khdrbs1⁻/⁻* NPCs-derived neurons display more complex morphology.

**Figure supplement 3.** High-throughput analyses of the transcriptomic of *Khdrbs1⁻/⁻*NPCs.

*figure supplement 1B*). Since ALDH1A3 is a multimeric protein and homodimerization domains are found also in the N-terminus of the protein, which is present in ALDH1A3△, we checked if ALDH1A3△ could contribute to ALDH activity in combination with the full-length protein. ALDH1A3△ efficiently co-immunoprecipitated with ALDH1A3. However, its co-expression with the full-length protein reduced ALDH activity in a dose-dependent manner (*Figure 6—figure supplement 1C,D*). These observations suggest that, even if expressed at low levels, ALDH1A3△ could not contribute to ALDH activity in NPCs, but it could eventually exert a dominant-negative effect.

In line with the reduced expression of full length ALDH1A3, ALDH activity was markedly decreased in *Khdrbs⁻/⁻* NPCs (*Figure 6B*). Furthermore, ALDH1A3 staining, which is restricted to the SOX2⁺ cells in VZ and SVZ of the developing cortex, was also significantly reduced in E13.5 knockout embryos with respect to wild-type littermates (*Figure 6C*). These results indicate that Sam68 promotes ALDH1A3 protein expression in NPCs.

The localization of ALDH1A3 suggests that its expression is restricted to RGCs. Moreover, we found that the full-length *Aldh1a3* transcript rapidly declined upon differentiation of wild-type NPCs in culture (*Figure 6D*). Importantly, this decline was accompanied by reduced Sam68 expression and increased expression of ALDH1A3△ (*Figure 6D*), indicating that alternative 3'-end processing of the *Aldh1a3* transcript is a physiological process of NPC differentiation controlled by fluctuations in Sam68 expression.

In agreement with a direct modulation of *Aldh1a3* expression by Sam68, we also observed a highly significant correlation between expression of these genes in post-natal brain, with highest levels of both transcripts in olfactory bulb and cerebellum and lowest expression in the hypothalamus (*Figure 6E*). Furthermore, expression of full-length *Aldh1a3* transcript was reduced in olfactory bulb and cerebellum of *Khdrbs1⁻/⁻* mice, while the ALE-containing shorter transcript was induced (*Figure 6F*). These results indicate that Sam68 is required to promote the expression and activity of ALDH1A3 through modulation of its pre-mRNA 3'-end processing in both embryonic and post-natal mouse brain.

## ALDH1A3 promotes glycolytic metabolism and self-renewal in embryonic NPCs

ALDH1A3 promotes self-renewal and clonogenic potential of glioma stem cells (*Mao et al., 2013*), suggesting that its reduced expression may account for the phenotype of *Khdrbs1⁻/⁻* NPCs. To test this hypothesis, we transfected *Khdrbs1⁻/⁻* NPCs with constructs expressing either GFP-ALDH1A3 or GFP alone as control. Notably, expression of GFP-ALDH1A3 almost completely rescued the growth and clonogenic defects of knockout NPCs (*Figure 7A–C*). In line with its negative effect of ALDH activity, expression of ALDH1A3△ in wild-type NPCs blocked proliferation and interfered with formation of neurospheres, even though the cells remained alive (*Figure 7—figure supplement 1*). Moreover, knockdown of endogenous ALDH1A3 in wild-type NPCs promoted neuronal differentiation to an extent similar to that observed in *Khdrbs1⁻/⁻* NPCs (*Figure 7D,E*). Collectively, these

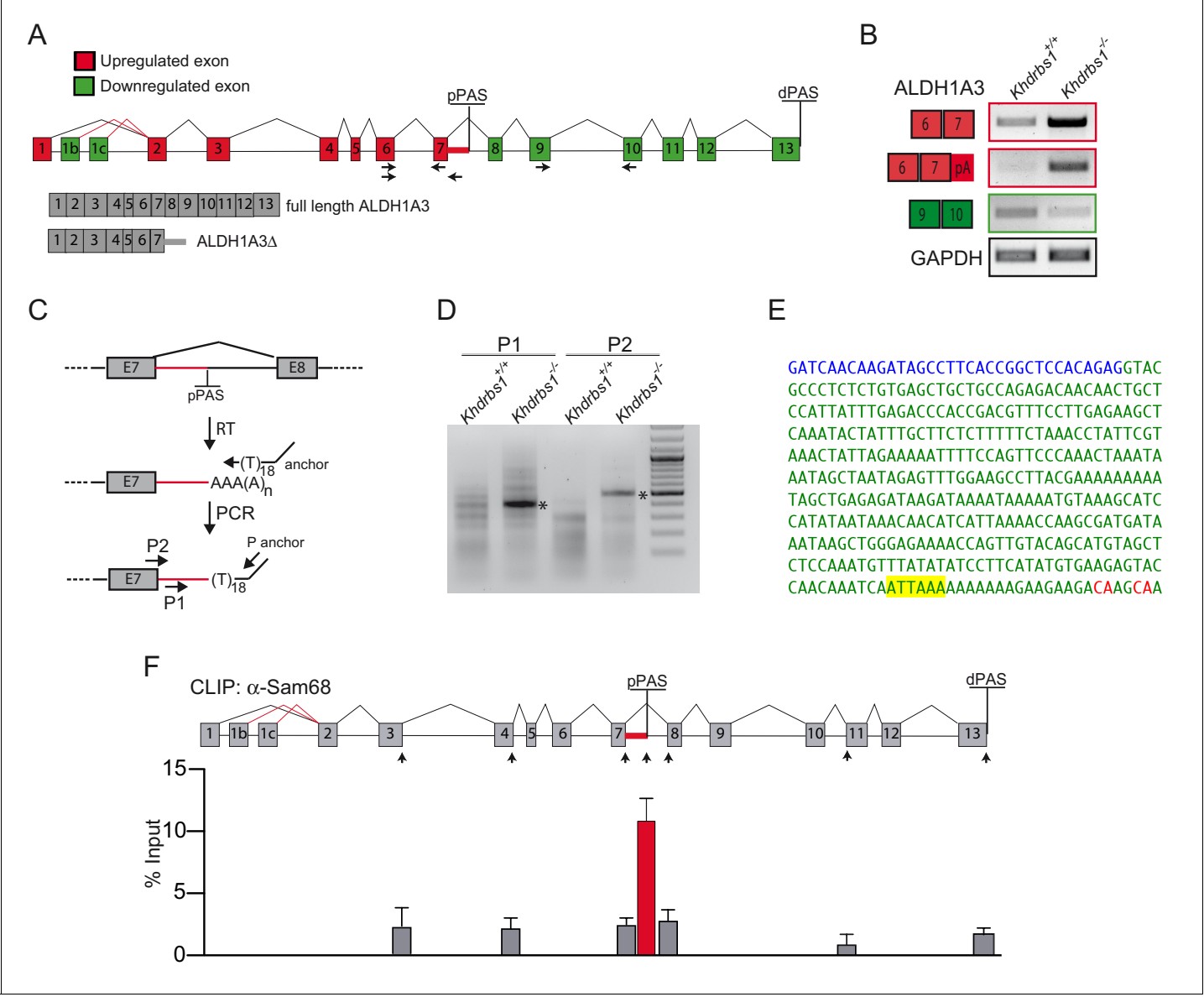

**Figure 5.** Sam68 regulates the alternative 3'-end processing of ALDH1A3 pre-mRNA. (A) Diagram of ALDH1A3 gene structure. Red and green squares represent exons, respectively, upregulated and downregulated in the microarray analysis. Proximal polyadenylation site (pPAS) and distal (dPAS) polyadenylation site are also represented. Black arrows indicate the regions were primers for RT-PCR validation were designed. (B) Conventional RT-PCR analysis for the validation of the canonical and alternative ALDH1A3 transcripts expressed in *Khdrbs1*[+/+] and *Khdrbs1*[-/-] NPCs. (C) Schematic representation of 3'-RACE PCR experimental design. Arrows indicate the position of the P1 and P2 forward primers used for the RACE experiment. (D) 3'-RACE PCR analysis of ALDH1A3 alternative transcripts expressed in *Khdrbs1*[+/+] and *Khdrbs1*[-/-] NPCs. Asterisks mark the bands that were sequenced to identify the pPAS. (E) Sequence obtained through DNA sequencing of bands amplified by 3'-RACE PCR. Blue and green letters represent exon and intron sequences, respectively. Exon (blue), intron (green), alternative PAS (highlighted in yellow) and the two putative cleavage sites (red) are indicated. (F) CLIP assay of Sam68 binding to the ALDH1A3 pre-mRNA. E13.5 mouse cortices were UV-crosslinked and immunoprecipitated with control IgGs or anti-Sam68 IgGs. The upper panel shows a schematic representation of ALDH1A3 gene structure and primers (black arrows) used in the assay. The bar graph shows qPCR signals amplified from the CLIP assays expressed as percentage of amplification in the input RNA.

The following figure supplement is available for figure 5:

**Figure supplement 1.** Sam68 regulates the alternative 3'-end processing of ALDH1A3 pre-mRNA.

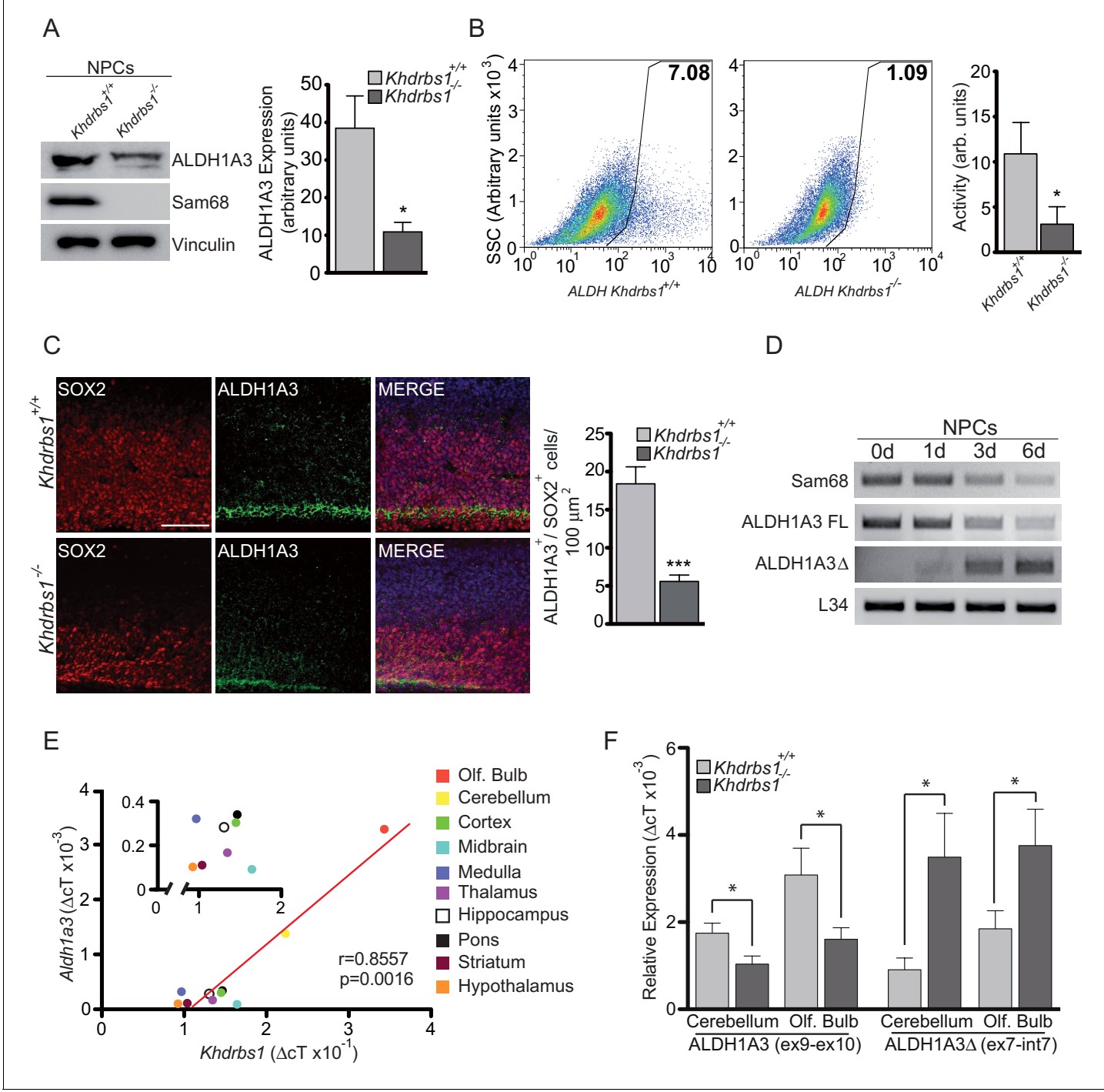

**Figure 6.** Sam68 promotes ALDH1A3 expression and activity. (**A**) Western blot analysis of the expression of ALDH1A3 and Sam68 in lysates from *Khdrbs1*$^{+/+}$ and *khdrbs1*$^{-/-}$ NPCs. Vinculin was used as loading control. The bar graph represent densitometric analyses of ALDH1A3 expression (mean±SD of 3 independent experiments; * p<0.05). (**B**) Representative flow cytometric analysis of ALDH activity performed on *Khdrbs1*$^{+/+}$ and *khdrbs1*$^{-/-}$ NPCs (left and center panel) by ALDEFLUOR assay. Gate was determined using NPCs treated with DEAB as negative control. SSC: side scatter value to measure cell complexity. Results (mean±SD; * p<0.05) of 4 independent experiments are reported in the bar graph on the right. (**C**) Horizontal sections of E13.5 *Khdrbs1*$^{+/+}$ and *khdrbs1*$^{-/-}$ cortex immunostained with ALDH1A3 and SOX2 antibodies. Scale bar 100 μm. Graph on the right represents counts of ALDH1A3$^+$/SOX2$^+$-positive cells. N = 3; *** p<0.001. (**D**) RT-PCR analysis of the expression of full length *Aldh1a3*, *ALDH1A3△*, and *Khdrbs1* in *Khdrbs1*$^{+/+}$ NPCs cultured under proliferating (0d) and differentiating conditions (1-6d). (**E**) Correlation analysis between the expression levels of *Khdrbs1* and full length *Aldh1a3* determined by qPCR analysis of RNA extracted from the indicated brain regions. Pearson's correlation test was used. Trend line is represented in red and the inset shows the magnification of the respective portion of the graph. (**F**) qPCR

*Figure 6 continued on next page*

*Figure 6 continued*

analysis of the expression of full length *Aldh1a3* (Ex 9–10) and ALDH1A3△ (Ex7– In7) in *Khdrbs1*[+/+] and *khdrbs1*[-/-] cerebellum and olfactory bulbs. N = 3; *p<0.05. (E–F) *Khdrbs1* and *Aldh1a3* relative expression was evaluated by △CT method using *L34* expression for normalization.
The following figure supplement is available for figure 6:

**Figure supplement 1.** Sam68 promotes ALDH1A3 expression and activity.

results support the notion that Sam68 promotes NPC proliferation and represses their differentiation by maintaining high levels of full-length ALDH1A3.

Stem cells often rely on anaerobic glycolysis for their proliferation and self-renewal (*Ito and Suda, 2014*) and high glycolytic activity is required to prevent precocious NPC differentiation in vivo (*Lange et al., 2016*). Since ALDH1A3 is involved in this metabolic pathway (*Mao et al., 2013*), we investigated whether Sam68 expression affects anaerobic glycolysis in NPCs. Lower accumulation of the end-product lactate in the culture medium indicated that glycolytic activity was significantly reduced in *Khdrbs1*[-/-] NPCs compared to wild-type cells (*Figure 7—figure supplement 2A*). A similar reduction in lactate production could be obtained by inhibiting ALDH activity using N,N-diethylaminobenzaldehyde (DEAB) (*Figure 7—figure supplement 2B*), a commonly used inhibitor of ALDH enzymes (*Morgan et al., 2015*). Furthermore, knockdown of *Aldh1a3* expression in wild type NPCs was sufficient to reduce anaerobic glycolysis to the levels of knockout cells, whereas overexpression of GFP-ALDH1A3 fully rescued lactate production in *Khdrbs1*[-/-] NPCs (*Figure 7F*). Notably, reduced glycolytic activity accompanied differentiation of NPCs in vitro (*Figure 7—figure supplement 2C*), concomitant with the decline in Sam68 and *Aldh1a3* expression (*Figure 6D*). These findings indicate that Sam68 maintains high levels of anaerobic glycolysis through modulation of ALDH1A3 protein expression.

Next, to determine whether inhibition of ALDH activity and glycolytic metabolism could recapitulate the phenotype of *Khdrbs1*[-/-] NPCs, we treated wild-type cells with 50 µM DEAB, a dose that lowered anaerobic glycolysis to the levels observed in knockout NPCs (*Figure 7—figure supplement 2B*). Strikingly, pharmacologic inhibition of ALDH activity in wild-type NPCs was sufficient to recapitulate the neurosphere growth (*Figure 7G*) and clonogenic defect (*Figure 7H*) of *Khdrbs1*[-/-] NPCs. These results strongly indicate that Sam68-dependent modulation of ALDH1A3 expression is required to maintain self-renewal capacity of NPCs through enhanced glycolytic metabolism (*Figure 8*).

## Discussion

The multifunctional RNA-binding protein Sam68 has been involved in several biological processes (*Bielli et al., 2011*; *Frisone et al., 2015*). Analysis of the mouse knockout model revealed that Sam68 is physiologically required for adipogenesis (*Richard et al., 2005*; *Huot et al., 2012*), male and female gametogenesis (*Paronetto et al., 2009*; *Bianchi et al., 2010*; *Paronetto et al., 2011*) and for regulation of synapse morphology and function in neurons (*Iijima et al., 2011*; *Klein et al., 2013*, *2015*). In most cases, its role was shown to require direct binding to select mRNAs and regulation of their splicing or translation. Notably, *Khdrbs1*[-/-] mice also displayed high perinatal lethality (*Richard et al., 2005*), but the biological processes and the genes regulated by Sam68 during embryonic development are completely unknown. Our studies have now unveiled an unpredicted function of Sam68 as key regulator of the balance between stem and differentiated cells in the developing cortex. Ablation of this function in vivo caused anticipated NPC differentiation into post-mitotic neurons and limited the expansion of the neocortex. These findings highlight a novel Sam68-dependent pathway required for stem cell fate during brain development.

Sam68 is strongly expressed in NPCs during stages of intense neurogenesis and cortical expansion (E13.5–E15.5). These processes rely on the dual ability of RGCs to divide symmetrically and amplify their number, or asymmetrically to produce IPCs and post-mitotic neurons (*Paridaen and Huttner, 2014*; *Taverna et al., 2014*). Knockout of Sam68 function significantly decreased proliferating NPCs (both PAX6-positive RGCs and TBR2-positive IPCs) at E13.5, while leading to a sharp increase in TBR1-positive post-mitotic neurons. Birth-dating experiments indicated that *Khdrbs1*[-/-]

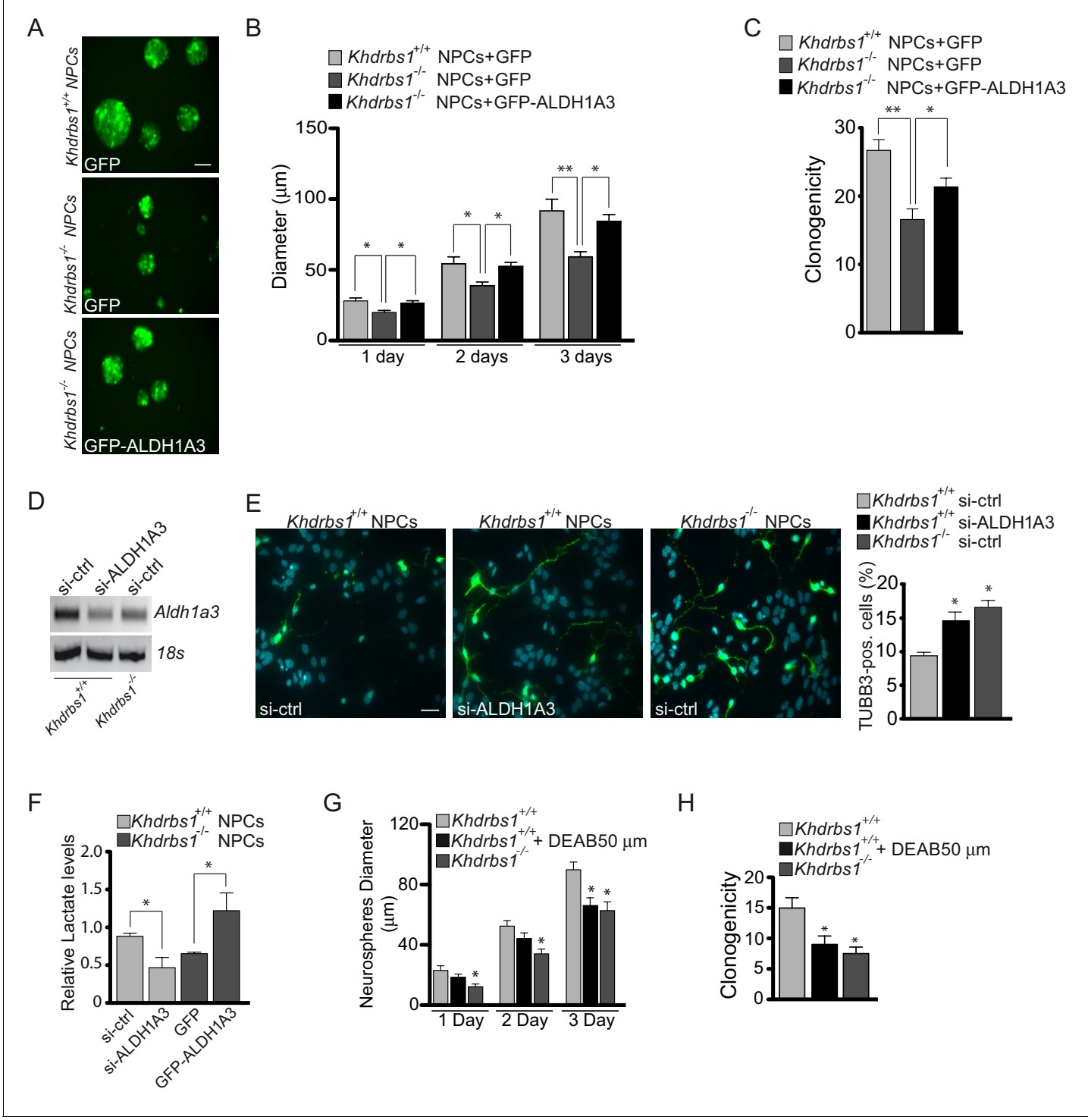

**Figure 7.** ALDH1A3-dependent glycolytic metabolism rescues the self-renewal potential of *Khdrbs1*[-/-] NPCs. (**A**) Representative images of *Khdrbs1*[+/+] NPCs transfected with GFP vector and *Khdrbs1*[-/-] NPCs transfected with GFP and GFP-ALDH1A3 vectors, cultured for 3 days in proliferating condition. Scale bar 50 µm. (**B**) Bar graph reporting the analysis of the diameter of neurospheres formed by transfected NPCs in 1–3 days of culture. N = 3; *p<0.05; **p<0.01. (**C**) Clonogenic assay of *Khdrbs1*[+/+] and *Khdrbs1*[-/-] NPCs transfected as indicated. N = 3; *p<0.05. (**D**) RT-PCR analysis of *Aldh1a3* expression in *Khdrbs1*[+/+] and *Khdrbs1*[-/-] NPCs transfected with si-CTRL or si-ALDH1A3 siRNAs. (**E**) Differentiation assay of *Khdrbs1*[+/+] and *Khdrbs1*[-/-] NPCs transfected with the same siRNAs used in (**D**). Images are representative of the third day of culture in differentiation condition. Scale bar 50 µm. Bar graph represent (mean±SD) measurement of number of TUBB3[+] cells. N = 3; * p<0.05 (**F**) Glycolytic activity measured by lactate accumulation in the medium of *Khdrbs1*[+/+] and *Khdrbs1*[-/-] NPCs transfected as indicated. (**G**) Bar graph reporting the analysis of the diameter of

*Figure 7 continued on next page*

*Figure 7 continued*

neurospheres formed by *Khdrbs1*[+/+] and *Khdrbs1*[-/-] NPCs cultured in proliferating condition for 3 days in the presence or absence of the ALDH inhibitor DEAB (50 μM). N = 3; *p<0.05. (**H**) Clonogenic assay of *Khdrbs1*[+/+] and *Khdrbs1*[-/-] NPCs cultured in proliferating condition in the presence or absence of 50 μM DEAB. N = 3; *p<0.05.

The following figure supplements are available for figure 7:

**Figure supplement 1.** ALDH1A3-dependent glycolytic metabolism rescues the self-renewal potential of *Khdrbs1*[-/-] NPCs.

**Figure supplement 2.** ALDH1A3-dependent glycolytic metabolism rescues the self-renewal potential of *Khdrbs1*[-/-] NPCs.

NPCs were prone to exit the cell cycle and differentiate into neurons. The consequence of this premature NPC loss was a reduced expansion of the cortex at E17.5, indicating that Sam68 is strictly required for the correct development of the mouse brain. Of note, the same loss of NPC stemness features could be recapitulated in vitro. First, Sam68 was strongly expressed in self-renewing E13.5

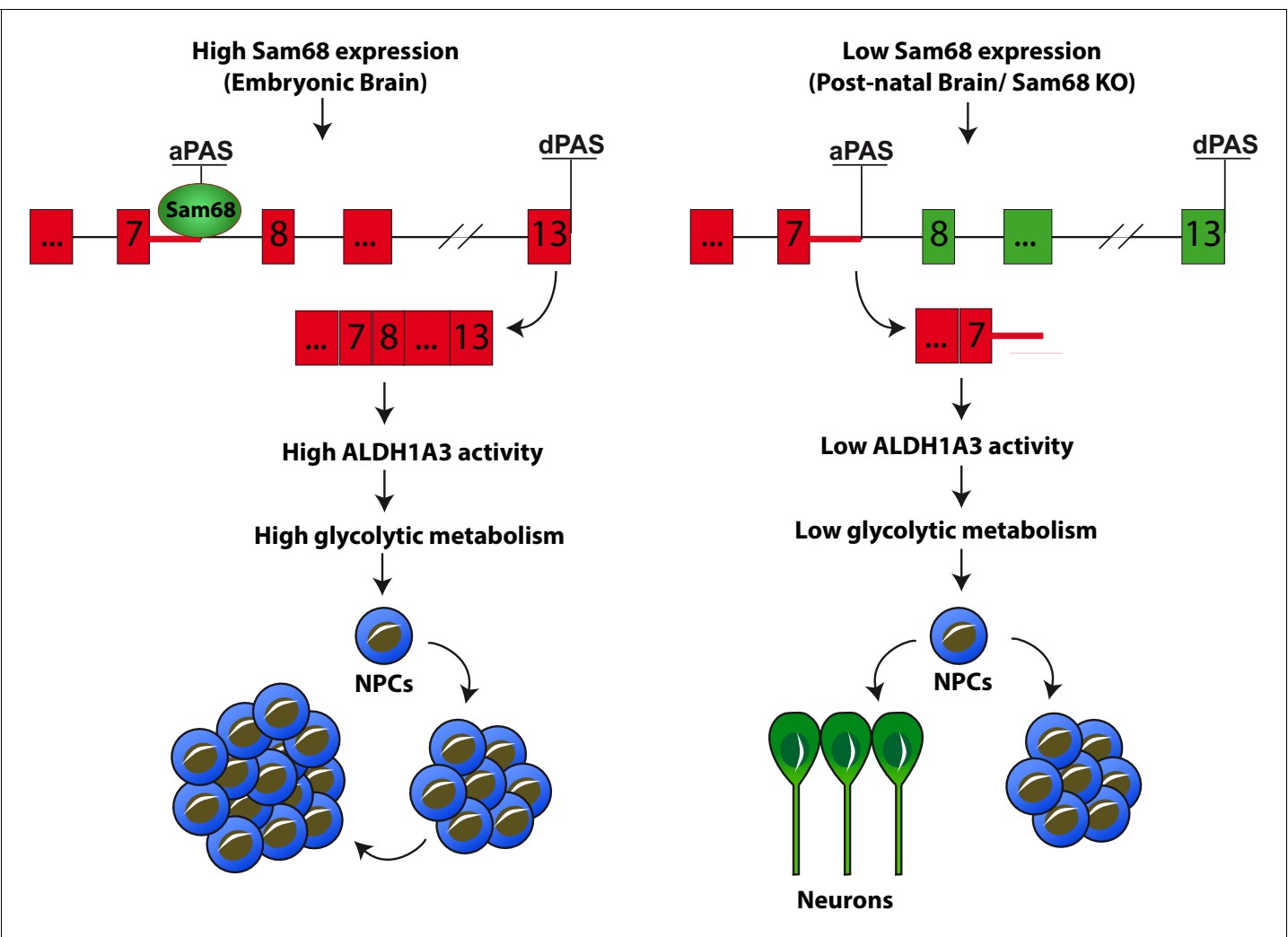

**Figure 8.** Sam68 modulates neurogenesis through regulating ALDH1A3 expression. Schematic model of the function of Sam68 in the modulation of NPCs stemness. High Sam68 expression in NPCs of the neocortex insures high expression of ALDH1A3 protein and anaerobic glycolysis, thus promoting NPC self-renewal and cortical expansion. Upon differentiation, Sam68 expression declines, causing premature termination of *Aldh1a3* transcription, reduced ALDH activity and glycolytic metabolism, thus enhancing NPC differentiation into post-mitotic neurons.

NPCs and its expression declined upon their differentiation. Second, *Khdrbs1*[-/-] NPCs were deficient in proliferation and clonogenicity while displaying enhanced tendency to differentiate. These observations indicate that Sam68 is required to maintain the proper balance between NPC self-renewal and differentiation and suggest that this protein supports stem-like properties in embryonic NPCs.

Analysis of splicing-sensitive microarrays identified a relatively small subset of genes that are modulated by Sam68 in NPCs. These genes were selectively enriched in functional categories typical of neurons and glial cells and the majority of them were upregulated in *Khdrbs1*[-/-] NPCs, in line with the role of Sam68 in the promotion of stem-like features. Among these targets, we focused on ALDH1A3, a metabolic enzyme that was identified as marker of cancer stem cells. Since ALDH1A3 supports the clonogenic and tumorigenic potential of cancer stem cells (*Duan et al., 2016*; *Mao et al., 2013*), we hypothesized that it might also promote stemness in NPCs and that Sam68 exerted its effects through modulation of ALDH1A3 expression. Indeed, *Khdrbs1*[-/-] NPCs displayed lower ALDH enzymatic activity and ALDH1A3 protein expression. Furthermore, by performing knockdown and gain-of-function studies, we demonstrated that regulation of ALDH1A3 expression accounts for many of the phenotypes of *Khdrbs1*[-/-] NPCs. Thus, our results link the physiological defect of knockout embryos to regulation of a specific target that promotes NPC self-renewal. Although it is generally accepted that pre-mRNA processing is a key regulatory step in brain development (*Raj and Blencowe, 2015*), few studies so far have directly linked these processes. Knockout of *Ptbp2* was shown to alter the dynamics of embryonic neurogenesis and splicing of several genes, but the specific target gene(s) of PTBP2 that are responsible for defective neurogenesis were not identified (*Licatalosi et al., 2012*). Likewise, the *Pnky* long noncoding RNA is required to expand the IPC population and it functions by interacting with PTBP1 and by regulating the expression and splicing of several neuronal genes (*Ramos et al., 2015*), but the specific targets involved in its neurogenic function are unknown. Our study extends beyond these correlations and establishes a direct link between Sam68 and ALDH1A3 functions in the maintenance of NPC self-renewal capacity during neurogenesis.

ALE selection is a prominent and unexpected pattern of regulation by Sam68 in NPCs. Mechanistically, we found that Sam68 directly binds to *Aldh1a3* pre-mRNA in proximity of a cryptic PAS located in the 5' portion of intron 7. Ablation of Sam68 unmasked this intronic PAS, causing alternative polyadenylation and premature termination of the *Aldh1a3* transcript. Notably, the same regulation of *Aldh1a3* alternative 3'-end processing occurred also in wild-type NPCs undergoing differentiation, concomitantly with the decline in expression of endogenous Sam68 and with acquisition of neuronal features. Thus, our results indicate that high levels of Sam68 are required to maintain *Aldh1a3* expression and activity by suppressing premature termination of transcription of its pre-mRNA (*Figure 8*). This function of Sam68 resembles that of the U1 snRNP, which prevents global recognition of a large number of cryptic intronic PASs, thus insuring proper transcription termination of pre-mRNAs in the last exon (*Kaida et al., 2010*; *Berg et al., 2012*). However, Sam68 appears to play a much more restricted role than U1 snRNP, by controlling 3'-end processing of only a limited subset of transcripts. It will be interesting to investigate whether or not Sam68 functionally interacts with U1snRNP in this regulation and why its expression affects the alternative 3'-end processing of only few transcripts. Nevertheless, the studies shown here unveil a previously unappreciated role of Sam68 in alternative polyadenylation with physiological relevance for NPC differentiation.

ALDH1A3 was shown to support clonogenicity and tumorigenicity of glioma stem cells by enhancing the glycolytic pathway (*Mao et al., 2013*). Notably, very recent studies demonstrated that anaerobic glycolysis is required for NPC self-renewal in vivo (*Lange et al., 2016*). We observed that lactate production through the glycolytic pathway was strongly reduced in *Khdrbs1*[-/-] NPCs. This phenotype could be recapitulated by *Aldh1a3* knockdown or ALDH pharmacological inhibition in wild-type NPCs, or by promoting NPC differentiation in culture. Furthermore, the clonogenic potential of both wild-type and knockout NPCs was tightly linked to the extent of glycolysis, as modulation of this metabolic pathway by up- or down-regulating ALDH1A3 activity paralleled their self-renewal potential. By linking ALDH1A3 activity to anaerobic glycolysis in NPCs, we identify a novel metabolic route required for maintenance of stemness in neural precursors that is strictly under the control of Sam68. Our work suggests that during neurogenesis, high levels of Sam68 expression are required for the correct processing of *Aldh1a3* pre-mRNA, thus insuring high ALDH1A3 activity and fueling of the glycolytic pathway. Conversely, a decline in Sam68 expression leads to reduced expression and

activity of ALDH1A3 and inhibits the glycolytic pathway, priming NPCs to enter the differentiation program (*Figure 8*). Importantly, the impaired balance between self-renewal and differentiation in *Khdrbs1*[-/-] mice results in depletion of the NPC pool and reduced expansion of the neocortex. Since altered neurogenesis and cortical development are associated to severe neuronal disorders like epilepsy, schizophrenia and autism (*Sun and Hevner, 2014*; *Fernández et al., 2016*), our findings suggest that fine-tuned regulation of Sam68 expression represents a safeguard mechanism during brain development.

In conclusion, our study identifies Sam68 as a novel regulator of neurogenesis and highlights an unprecedented link between Sam68, ALDH1A3 and the glycolytic pathway that supports maintenance of the NPC pool in the embryonic cortex.

## Materials and methods

### Ethics statement
Animal experiments were performed according to protocol number 809_2015PR, following the Institutional guidelines of the Fondazione Santa Lucia and the approval of the Ethical Committee.

### Immunofluorescence analysis of neocortex development
E12.5, E13.5 and E17.5 pregnant mice were treated with 90 mg/kg BrdU (Sigma-Aldrich, St. Louis, MO). After 2 days (E12.5 mice) or 2 hr (E13.5 and E17.5 mice), mice were sacrificed and whole embryo (E12.5 and E13.5) or embryonic brains (E17.5) were collected and fixed in formaldehyde 4%/PBS (v/v) for 8 hr, transferred for 24 hr to a Sucrose/PBS 30% (w/v) solution and then frozen to −80°C. Brains were embedded in Tissue-Tek OCT before collecting the cryosections (40–45 µm thick) in PBS 0.01% sodium azide (w/v). Sections were treated with 0.1 N HCl for 20 min at 37° C and then in 1 M sodium borate for 20 min at room temperature (RT). Floating sections were incubated overnight at 4°C with the following antibodies: goat anti-SOX2 (1:300; Santa Cruz, Santa Cruz, CA, RRID:AB_2286684), rabbit anti-TBR1 (1:300; Abcam, Cambridge, MA, RRID:AB_2200219) or anti-TBR2 (1: 300; Abcam, RRID:AB_778267), rat anti-BrdU (1:400; AbDSerotech, Hercules, CA, RRID:AB_323427), rabbit anti-PAX6 (1:100; Covance, Princeton, NJ, RRID:AB_2313780), rabbit anti-Ki67 (1:200; LabVision, Waltham, MA, RRID:AB_2335745), rabbit anti-Sam68 (1:1000; Santa Cruz, RRID:AB_631869) and rabbit anti-ALDH1A3 (1:100; Sigma-Aldrich, RRID:AB_10607146). Secondary antibodies (Jackson ImmunoResearch, West Grove, PA) were used as follows: Cy3-conjugated donkey anti-rat, 1:300; Cy2-conjugated donkey anti-rabbit, 1:300; Alexa647-conjugated donkey anti-goat, 1:300. Images were collected by laser-scanning confocal microscopy using a TCS SP5 microscope (LEICA microsystem, Switzerland) and elaborated with Photoshop (Adobe, San Jose, CA).

### NPCs isolation, culture and immunofluorescence analysis
NPCs were isolated from *Khdrbs1*[+/+] and *Khdrbs1*[-/-] C57/BL6 (Charles River Laboratories, RRID:MGI: 3696370) E13.5 (*Bard et al., 1998*) mouse embryos as previously reported (*Compagnucci et al., 2013*). Briefly, after olfactory bulbs, ganglionic eminences and meninges removal, embryonic cortices were enzymatically digested with Papain 30 U/ml (Sigma-Aldrich), 0.24 mg/ml L-Cysteine (Sigma-Aldrich), 40 mg/ml DNaseI (Sigma-Aldrich) dissolved in MEM (Sigma-Aldrich) for 10 min at 37°C to obtain cell suspensions. After centrifugation, cells were seeded in neurosphere medium consisting of DMEM:F12 (1:1) (Sigma-Aldrich) containing 0.2 mg/ml L-glutamine (Sigma-Aldrich), B27 (1 ml/50 ml, Gibco), penicillin (100 U/ml), streptomycin (100 mg/ml) (all from Lonza, Switzerland), and supplemented with 20 ng/ml epidermal growth factor (EGF) (Peprotech, United Kingdom) and 20 ng/ml basic fibroblast growth factor (bFGF) (Peprotech), and passaged every 4–5 days. For clonal analysis, 5000 NPCs were plated in 35-mm wells for each experimental point. After 5 days of culture in proliferating conditions, neurosphere number was evaluated and NPC clonogenity was calculated as ratio between plated cells and neurospheres formed, expressed as percentage. For differentiation assays, 20,000 NPCs/well were plated in 4-well dishes, pre-coated with poly-ornithine (Sigma-Aldrich) in $H_2O$ and with laminin-1 (Sigma-Aldrich) in PBS for 1 hr each at 37°C, in neurosphere medium containing 1% v/v fetal bovine serum (FBS) (Gibco, United Kingdom) and incubated in a humidified atmosphere with 6% $CO_2$ at 37uC for 1 to 6 days.

For immunofluorescence analysis, NPCs cultured in differentiating condition were fixed with 4% (v/v) formaldehyde (Sigma-Aldrich) and permeabilized with 0.1% Triton X-100 in PBS, supplemented with 1% BSA. Primary antibodies (1 hr at RT): mouse anti-Nestin (1:1000; Millipore, Billerica, MA, RRID:AB_94911), rabbit anti-PAX6 (1:500; Covance, RRID:AB_2313780), rabbit anti-GFP (1:300; Abcam, RRID:AB_303395), mouse anti-GFP (1:300; Santa Cruz, RRID:AB_627695), mouse anti-TUBB3 (1:300; Sigma-Aldrich, RRID:AB_1841228), chicken anti-MBP (1:50; Millipore, RRID:AB_ 2140366). Specimens were then incubated with Cy3- (1:500) and FITC-conjugated (1:250) secondary antibodies (Jackson ImmunoResearch) for 1 hr at RT and with Hoechst (Invitrogen, United Kingdom) for 15 min and preserved using Prolong Gold mounting solution (Invitrogen). 10 randomly taken fields for each sample were taken using a DMI6000B inverted microscope (LEICA Geosystems) equipped with a Pan-Neofluar 20× /0.75 objective lens. Data are represented as percentage of positive cells/total cells (evaluated by the number of total nuclei).

## NPC transfection

For transfection, NPCs were electroporated using AMAXA nucleoflector device II and AMAXA mouse NSC nucleoflector kit (Lonza), following manufacturer instructions. Briefly, $5 \times 10^6$ NSCs were centrifuged and resuspended in 100 µl of P3 primary solution with 5 µg of GFP or ALDH1A3-GFP vectors or 300 nM of si-ctrl or a mix of 3 siALDH1A3 siRNAs. Electroporated NPCs were resuspended in 500 µl of pre-warmed medium, centrifuged at 1300 rpm for 5 min at RT and resuspended in NPC proliferation medium. siRNAs sequence is reported in *Supplementary file 1*.

## RNA isolation, RT-PCR and RT-qPCR

Total RNA was extracted from NPCs or brain tissues using Trizol reagent (Invitrogen) following manufacturer instructions. 1 µg of RNA was retrotranscribed using M-MLV reverse transcriptase (Invitrogen) and used in RT-PCR experiments using GoTaq Flexi DNA Polymerase (Promega Corporation, Madison, WI) or in quantitative RT-PCR (qPCR) experiments, using SIBR green PCR master mix (Applied Biosystem) following manufacturers instructions. All primers used in RT-PCR and qPCR experiments are reported in *Supplementary file 1*. *L34* gene expression was for normalization in qPCR experiments.

## 3' RACE PCR experiments

For 3' RACE PCR, 2 µg of total RNA isolated from *Sam68*$^{+/+}$ and *Sam68*$^{-/-}$ NPCs was used for retrotranscription with 0.5 µg of oligo dT18XbaI-KpnI-BamHI primer, comprising an oligo-dT tail followed by an anchor sequence (see *Supplementary file 1*). PolyA enriched/anchor tagged cDNA was subsequently used in RT-PCR experiments using gene-specific forward primers in presence of anchor reverse primer (see *Supplementary file 1*). PCR products were resolved in agarose gel and bands of interest were purified and sequenced.

## Immunoblotting

NPCs or brain tissues were lysed in 50 mM Tris–HCl, pH 7.4; 100 mM NaCl; 1 mM MgCl2; 0.1m MCaCl2; 1% NP-40; 0.5% sodium deoxycholate; 0.1% SDS, protease inhibitor cocktail, plus phosphatase and protease inhibitors (Sigma-Aldrich). After protein separation by SDS–PAGE and transfer to polyvinyl difluoride (PVDF) membranes (Amersham, United Kingdom), the following antibodies were incubated in 5% BSA in TBS-0.1% Tween buffer: rabbit anti-Sam68 (1:1000, Santa Cruz, RRID: AB_631869), rabbit anti-SOX2 (1:1000, Millipore, RRID:AB_2286686), rabbit anti-ALDH1A3 (1:1000, Abgent, San Diego, CA, RRID:AB_2224040), mouse anti-GAPDH (1:1000, Santa Cruz, RRID:AB_ 627679), mouse anti-Vinculin (1:1000, Sigma-Aldrich, RRID:AB_10603627). Signals were detected by enhanced chemiluminescence (ECL) (Biorad).

## Cloning of full-length ALDH1A3 and ALDH1A3△

Total RNA was extracted and retrotranscribed as reported above and cDNA was used to clone ALDH1A3 FL and △ isoforms in pEGFP-C3, p3xFlag-CMV or pCI vectors. Inserts were amplified using primers containing HindIII (forward) or PstI (reverse) recognition sites in pEGFP-C3, p3xFlag-CMV vectors cloning and EcorI (forward) or NotI (reverse) recognition sites in pCI vector cloning and High fidelity Phusion DNA Polymerase (Thermo Scientific, Waltham, MA), following manufacturer

instructions. All products were verified by sequencing. Sequence of primers used is reported in *Supplementary file 1*.

## ALDH activity assay

ALDH activity using the ALDEFLUOR assay kit (Stem cells, Canada). $1 \times 10^6$ NPCs were resuspended in Aldefluor assay buffer containing BAAA (1 μmol/l) for 60 min and maintained on ice. For each sample, the same number of cells was treated with 50 mmol/l of diethylaminobenzaldehyde (DEAB) ALDH inhibitor, to set the negative control gate. Flow cytometric analyses were conducted by exciting samples at 488 nm and detecting emission light using a standard fluorescein isothiocyanate (FITC) channel.

## Assay of glycolytic metabolism

To evaluate glycolytic metabolism, a glycolysis cell-based assay kit (Sigma-Aldrich) was used. $1 \times 10^4$ NPCs were seeded in a 96-well plate with 200 μL of neurosphere medium and cultured overnight. 25 μl of supernatant from cultured cell plates and 25 μl of assay buffer were mixed together and then 50 μL of reaction solution were added. 6 samples were used to set up the standard curve by reading absorbance of each well at 490 nm was read and L-lactate concentrations of each sample were calculated using the corrected absorbance of each sample and interpolating it on the standard curve.

## CLIP assays

CLIP assays were performed as previously described (*Wang et al., 2009*), with some modification. E13.5 cortices were dissected, freed from meninges and dissociated mechanically. After irradiation with UV light on ice (400 mJ/cm²), samples were incubated with lysis buffer (50 mM Tris–HCl, pH 7.4; 100 mM NaCl; 1 mM MgCl2; 0.1mMCaCl2; 1% NP-40; 0.5% sodium deoxycholate; 0.1% SDS, protease inhibitor cocktail, RNase inhibitor), briefly sonicated and treated with DNase-RNase free for 3 min at 37°C. After centrifugation at 15,000xg for 3 min at 4°C, 500 μg of extract was treated with Proteinase K (PK) for 30 min at 37°C and RNA was purified by standard procedure (input) or diluted to 1 ml with lysis buffer for immunoprecipitation with anti-Sam68 (Santa Cruz, RRID:AB_631869) or control rabbit IgGs (Sigma-Aldrich) as negative control, in presence of protein-G magnetic dynabeads (Life Technologies, United Kingdom) and 10 μl RNase I (1:1000, Ambion, Waltham, MA), for 2 hr at 4°C under rotation. After stringent washes in high salt, dynabeads were resuspended in PK buffer. 10% of each sample was kept as control of immunoprecipitation and the rest was treated with 50 μg PK for 1 hr at 55°C. RNA was then isolated and used for qPCR analysis.

## Microarray analysis

Total RNA was isolated from independent NPCs obtained from 3 wild type and 3 knockout E13.5 mouse embryos. RNA was hybridized to GeneChip Mouse Exon 1.0 ST Arrays (Affymetrix, Santa Clara, CA) at Genosplice, Paris. Bioinformatics analyses for gene expression and splicing index was performed at Genosplice using the EASANA software and FAST-DB as reference database.

## Statistics

All data are expressed as mean ± SD. Student's t-test and ANOVA were performed using Graphpad Prism software (RRID:SCR_002798).

## Acknowledgements

The authors wish to thank Dr. Maria Blaire Bustamante for help with microarrays, Dr. Vittoria Pagliarini for help with CLIP assays, and Dr. Pierre de la Grange (Genosplice, Paris) for microarray analyses. This work was supported by grants from Telethon (GGP 12189; GGP 14095), Associazione Italiana Ricerca sul Cancro (AIRC; IG14581), Muscular Dystrophy Association (MDA), and from Ministry of Health 'Ricerca Corrente' and '5x1000 Anno 2014' to Fondazione Santa Lucia. PGL was partly supported by a scholarship from Fondazione Umberto Veronesi.

## Additional information

### Funding

| Funder | Grant reference number | Author |
|---|---|---|
| Telethon | GGP14095 | Claudio Sette |
| Associazione Italiana Ricerca sul Cancro | IG14581 | Claudio Sette |
| Muscular Dystrophy Association | | Claudio Sette |
| Telethon | GGP 12189 | Claudio Sette |

The funders had no role in study design, data collection and interpretation, or the decision to submit the work for publication.

### Author contributions

PLR, CS, Conception and design, Acquisition of data, Analysis and interpretation of data, Drafting or revising the article; PB, CC, EC, Acquisition of data, Analysis and interpretation of data, Contributed unpublished essential data or reagents; EV, Acquisition of data, Analysis and interpretation of data, Drafting or revising the article; SFV, Conception and design, Acquisition of data, Analysis and interpretation of data

### Author ORCIDs

Claudio Sette, http://orcid.org/0000-0003-2864-8266

### Ethics

Animal experimentation: This study was performed in strict accordance with the recommendations in the Guide for the Care and Use of Laboratory Animals of the Italian Ministry of Health. All of the animals were handled according to approved institutional animal care and use committee of the University ofRome Tor Vergata. The protocol was approved by the Committee on the Ethics of Animal Experiments of the University of Rome Tor Vergata. Every effort was made to minimize suffering of mice.

## Additional files

### Supplementary files

• Supplementary file 1. List of the oligonucleotides used as PCR primers and siRNAs in the study.

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
