## [Decision Letter]

Thank you for submitting your article "SAM68 promotes self-renewal and glycolysis in neural progenitor cells by modulating ALDH1A3 pre-mRNA 3'-end processing" for consideration by *eLife*. Your article has been reviewed by two peer reviewers, Stéphane Richard (Reviewer #1) and Emanuele Buratti (Reviewer #2), and the evaluation has been overseen by Marianne Bronner as the Senior Editor and Reviewing Editor.

The reviewers have discussed the reviews with one another and the Reviewing Editor has drafted this decision to help you prepare a revised submission.

Summary:

La Rosa et al. characterize the role of the KH-type RNA binding protein Sam68 in embryonic neurogenesis. They show that the ablation of Sam68depletes the neural stem cell pool, as it promotes their cell cycle exit and subsequent differentiation into neurons (and also oligodendrocytes in vitro). Interestingly, they observe a significant increase in TBR1^+^ neurons in *Sam68*^-/-^ mice. La Rosa et al., identify many alternative spliced events using alternative splice sensitive exon arrays and focus their attention on ALDH1A3. They show that Sam68 binds a cryptic intronic poly (A) signal and prevents its usage. The loss of Sam68 uses this cryptic signal and generates a truncated non-functional ALDH1A3 enzyme. They argue that the phenotype of the NSCs is due mainly to the loss of ALDH1A3 activity, as restoring its activity rescues the diameter of the Sam68-deficient neurospheres and their clonogenicity. Furthermore, depletion of wild type NPCs with siALDH1A3 also leads to an increase in neuronal differentiation as assessed by the TUBB3 marker. Finally, the authors demonstrate that high ALDH1A3 expression and activity are required for the high glycolytic metabolism necessary for maintaining the stem cell pool of the developing cortex. Thus La Rosa et al. convincingly show that it is mainly the lack of ALDH1A3 in a Sam68-dependent manner that contributes to the early cell cycle exit to promote neural differentiation of certain neurons.

Essential revisions:

1) Figure 1: the authors should show immunofluorescence staining for E13.5 cortex from *Sam68*^-/-^ mice to see whether Sam68 depletion affects the overall number of *Sox2*^+^ cells at the peak of neurogenesis. Figure 1: Could be improved and quantified with real-time PCR.

2) Figure 3: the authors demonstrate that cortical expansion is reduced in *Sam68*^-/-^ mice compared to wild type mice. When assessing such a measurement, do the authors account for the size and sex of these mice? Particularly since *Sam68*^-/-^ mice are known to be smaller in size than *Sam68*^+/+^ mice at that age?

3) Figure 6: the authors show that full length ALDH1A3 protein expression is decreased in *Sam68*^-/-^ NPCs; however, this seems to contradict the data in Figure 4—figure supplement 3, where ALDH1A3 transcript is shown to be increased in *Sam68*^-/-^ NPCs. Is the increase in transcript level specific to the ALDH1A3Δ isoform? Could the authors provide an explanation for this discrepancy?

4) Figure 6: the authors should show the quantification of *Sox2*^+^/ALDH1A3^+^ cells.

5) The experiments are all RT-PCR based (e.g. Figure 5—figure supplement 1) where a negative result cannot really represent proof of absence. To rule out whether there is no accumulation of intron 7 containing pre-mRNA, the authors should perform Northern blot analysis.

6) Using CLIP analysis, the authors showed that Sam68 binds this intron 7 region. It is not clear, however, why Sam68 was not observed to bind the dPAS in exon 13 of the ALDH1A3 gene. Does it have a sequence that significantly differs from AAUAAA or AUUAAA?

7) The authors found that ALDH1A3 protein expression was decreased. The authors showed that the truncated form was not expressed but upon ectopic expression, this isoform appeared to be enzymatically inactive. It would be interesting to know whether this truncated isoform is actually never translated or degraded immediately due to low stability. Have the authors tried to treat the *Sam68*^-/-^ cells with autophagy or proteasome inhibitors to see if this truncated isoform cab be actually visualized in western blots?

8) Finally, the authors show that overexpression of GFP-ALDH1A3 can rescue the growth and clonogenic defects in *Sam68*^-/-^ cells. These experiments are convincing but should also be repeated by GFP-ALDH1A3 truncated isoform may not under any circumstance rescue the *Sam68*^-/-^ cells.

---

## [Author Response]

*[…] Essential revisions:*

*1) Figure 1: the authors should show immunofluorescence staining for E13.5 cortex from Sam68^-/-^ mice to see whether Sam68 depletion affects the overall number of Sox2^+^ cells at the peak of neurogenesis. Figure 1: Could be improved and quantified with real-time PCR.*

As suggested, we now show counts of *Sox2*^+^ cells and document a significant reduction in the Sam68 knockout embryos, consistently with the reduction in proliferating BrdU-positive NPCs. This result is now shown in the new Figure 2—figure supplement 1.

Regarding Figure 1, we have now quantified the changes in gene expression for *Sam68, Sox2* and *Tubb3* by qPCR, confirming the results of conventional PCR. The data are shown in the new panel G.

*2) Figure 3: the authors demonstrate that cortical expansion is reduced in Sam68^-/-^ mice compared to wild type mice. When assessing such a measurement, do the authors account for the size and sex of these mice? Particularly since Sam68^-/-^ mice are known to be smaller in size than Sam68^+/+^ mice at that age?*

In all cases, we used female embryos, thus avoiding possible biases between sexes. We do not see significant differences in size between wt and KO embryos from E13.5 to E17.5, and genotyping is the only way to distinguish them. We now report the weight of embryos in support of this evidence in the new Figure 3—figure supplement 1.

*3) Figure 6: the authors show that full length ALDH1A3 protein expression is decreased in Sam68^-/-^ NPCs; however, this seems to contradict the data in Figure 4—figure supplement 3, where ALDH1A3 transcript is shown to be increased in Sam68^-/-^ NPCs. Is the increase in transcript level specific to the ALDH1A3Δ isoform? Could the authors provide an explanation for this discrepancy?*

Yes, as the reviewers point out, this apparent discrepancy is due to the fact that the 5’ region of ALDH1A3 (corresponding to expression of the ALDH1A3△ variant) appears up-regulated in the exon array whereas the 3’ region, necessary for the full length ALDH1A3 protein expression, is down-regulated. This is better illustrated in Figure 5 and its supplement, where we show by conventional and qPCR the validation of the array data. Thus, the increase in ALDH1A3 expression shown in Figure 4—figure supplement 3 corresponds to the ALDH1A3Δ isoform (for validation we used primers in exons 6-7 as in Figure 5—figure supplement 1). We now specify this better in the figure to avoid generating confusion in the readers.

*4) Figure 6: the authors should show the quantification of Sox2^+^/ALDH1A3^+^ cells.*

We now show this quantification in the new Figure 6.

*5) The experiments are all RT-PCR based (e.g. Figure 5—figure supplement 1) where a negative result cannot really represent proof of absence. To rule out whether there is no accumulation of intron 7 containing pre-mRNA, the authors should perform Northern blot analysis.*

We agree with the reviewer that a negative result is not conclusive. Thus, instead of performing a Northern blot, we designed a different experiment to discriminate between pre-mRNA and mature mRNA: *Sam68*^-/-^ NPCs were incubated with DRB for 2 and 6 hours to block RNA transcription. Under these conditions, pre-mRNAs, which are transient intermediates, rapidly decay, whereas mature transcripts should be more stable. Indeed, we observed that transcripts containing exon 3-intron 3 and intron 8-exon 8 sequences were present at time 0 and rapidly disappeared upon incubation with DRB, indicative of pre-mRNA amplification. By contrast, the amplicon corresponding to exon 7-intron 7 sequences (i.e. ALDH1A3△ transcript) was stable up to 6 hours of incubation. Notably, in this experiment all amplicons were of the same size, ruling out a possible bias related PCR efficiency. These results (now shown in Figure 5—figure supplement 1) confirm by another means the presence of an alternative mRNA originating from premature termination of the ALDH1A3 transcript in *Sam68*^-/-^ NPCs.

*6) Using CLIP analysis, the authors showed that Sam68 binds this intron 7 region. It is not clear, however, why Sam68 was not observed to bind the dPAS in exon 13 of the ALDH1A3 gene. Does it have a sequence that significantly differs from AAUAAA or AUUAAA?*

The canonical polyA site in exon 13 contains the most common polyA site sequence: AAUAAA. So, in theory it does not show a sequence bias for SAM68 recognition. One potential bias is the position of the polyA site: internal intron versus last exon. We performed a high-throughput analysis of Sam68-regulated splicing events in male germ cells and found that many intronic polyA sites are regulated by Sam68. It is possible that Sam68 function requires additional splicing proteins that associate with it but are more enriched within the gene body than at the cleavage site. We are currently investigating this possibility at mechanistic level, but do not have conclusive data yet to be added to this study.

*7) The authors found that ALDH1A3 protein expression was decreased. The authors showed that the truncated form was not expressed but upon ectopic expression, this isoform appeared to be enzymatically inactive. It would be interesting to know whether this truncated isoform is actually never translated or degraded immediately due to low stability. Have the authors tried to treat the Sam68^-/-^ cells with autophagy or proteasome inhibitors to see if this truncated isoform cab be actually visualized in western blots?*

We have now performed the requested experiment. MG132 did not lead to any accumulation of a truncated ALDH1A3 protein. Chloroquine caused a very mild accumulation of a protein product detected by the antibody, but slightly smaller than what expected by the apparent molecular weight of the recombinant ALDH1A3△. Positive controls were included to show their accumulation upon treatment: p53 for inhibition of proteosomal degradation and LC3 for inhibition of the autophagic flux. Thus, even if expressed at very low levels, ALDH1A3△ appears very unstable and eliminated through autophagy. Nevertheless, we tested its possible functional contribution to ALDH activity and NPC stemness (see point 8).

*8) Finally, the authors show that overexpression of GFP-ALDH1A3 can rescue the growth and clonogenic defects in Sam68^-/-^ cells. These experiments are convincing but should also be repeated by GFP-ALDH1A3 truncated isoform may not under any circumstance rescue the Sam68^-/-^ cells.*

We have now tested the possibility that ALDH1A3△ contributes to ALDH activity and neurogenic potential of NPCs. First, enzymatically active ALDH1A3 is known to form a tetramer, and homodimerization domains are found also in the N-terminus of the protein present in ALDH1A3△. Consistently, we found that ALDH1A3△ forms a complex with full length ALDH1A3 in co-immunoprecipitation experiments in HEK293 cells. However, its co-expression with full length ALDH1A3 partially reduced ALDH activity in a dose-dependent manner (new Figure 6—figure supplement 1). This experiment suggests that, if expressed, ALDH1A3△ could behave as dominant-negative and interfere with the enzymatic activity of the full length protein. In line with this possibility, we also show that forced expression of ALDH1A3△ in NPCs completely blocks their proliferation and interferes with formation of the neurospheres, even though the cells remain alive in the time-frame of the experiment, as they are still visible and express GFP 3-5 days after transfection. This result is now shown in the new Figure 7—figure supplement 1.